

# Estimating snow depth on Arctic sea ice based on reanalysis reconstruction and particle filter assimilation

Haili Li[1,2,3], Chang-Qing Ke[1,2,3], Qinghui Zhu[1,2,3], Xiaoyi Shen[1,2,3]

[1] Jiangsu Provincial Key Laboratory of Geographic Information Science and Technology, Key Laboratory for Land Satellite
Remote Sensing Applications of Ministry of Natural Resources, School of Geography and Ocean Science, Nanjing University, Nanjing, 210023, China
[2] Collaborative Innovation Center of Novel Software Technology and Industrialization, Nanjing, 210023, China
[3] Collaborative Innovation Center of South China Sea Studies, Nanjing, 210023, China

*Correspondence to*: Chang-Qing Ke (kecq@nju.edu.cn)

**Abstract.** The snow depth, an essential metric of snowpacks, can modulate sea ice changes and is a necessary input parameter to obtain altimeter-derived sea ice thickness values. In this study, we propose an innovative snow depth retrieval method with the improved NASA Eulerian Snow on Sea Ice Model (INESOSIM) and the particle filter (PF) approach, namely, INESOSIM-PF. Then, we generate daily snow depth estimates with INESOSIM-PF from 2012 to 2020 at a 50-km resolution. With the use of Operation IceBridge (OIB) data, it can be revealed that compared to the NESOSIM-estimated

snow depth, the INESOSIM-PF-estimated snow depth is greatly improved, with a root mean square error (RMSE) decrease of 17.97% (RMSE: 6.73 cm) and a correlation coefficient increase of 11.85% (r: 0.71). The INESOSIM-PF-estimated snow depth is close to the satellite-derived snow depth, which is applied in data assimilation. With the use of Multidisciplinary Drifting Observatory for the Study of Arctic Climate (MOSAiC) snow buoy data, it can be verified that INESOSIM-PF performs well in the Central Arctic with an RMSE of 9.23 cm. INESOSIM-PF is robust and the snow depth determined with

INESOSIM-PF is less influenced by input parameters with a snow depth uncertainty of 0.74 cm. The variations in the monthly and seasonal snow depth estimates retrieved from INESOSIM-PF agree well with those in the estimates retrieved from two other existing algorithms. Based on the presented snow depth estimates, we can retrieve the sea ice thickness and perform long-term snow depth and sea ice analysis. Snow depth estimates improve the understanding of Arctic environmental change and promote the future development of sea ice models.

## 1 Introduction

In the Arctic, snowpack on sea ice modulates variations in sea ice through the snow depth, density and distribution (Webster et al., 2014), playing an important role in Arctic hydrology, energy balance and climate (Serreze et al., 2006; Perovich and Polashenski, 2012; Handorf et al., 2015). Rapid sea ice change affects Arctic amplification (Screen and Simmonds, 2010; Screen and Francis, 2016). Arctic amplification is notable in areas where sea ice decreases sharply (Dai et al., 2019). The

high surface albedo of a snowpack (Perovich et al., 2002) limits solar radiation absorption, and its low thermal conductivity



(Sturm et al., 2002) inhibits heat transfer from the atmosphere to sea ice. The winter insulation effect of snowpacks on sea ice decelerates sea ice growth, and spring-summer snowpacks on sea ice impede sea ice melting. Meltwater originating from thin snow facilitates the formation of melt ponds, and the low reflectivity of water absorbs much solar radiation and thus promotes sea ice melt (Eicken et al., 2002; Petrich et al., 2012). The snow depth in most Arctic regions exhibits a negative

trend during the cold season (Stroeve et al., 2020). The spring snow depth atop sea ice decreases at a rate of -0.29 cm/year (Webster et al., 2014). Compared to data pertaining to 1937 and 1954–1991 retrieved from Soviet drifting stations (35.1±9.4 cm), the average snow depth (2009–2013) decreased to 22.2±1.9 cm, while the snow depth changed from $32.8 \pm 9.4$ cm to $14.5 \pm 1.9$ cm in the Beaufort and Chukchi seas, declining $56 \pm 33\%$ (Webster et al., 2014).

In the early days, the snow depth in local regions could be obtained through stations or buoys (Haas et al., 2017; Nicolaus,

2021). Subsequently, the popularity of ship-based underway observations and aerial surveys (Uto et al., 2006; Brucker and Markus, 2013) provided an effective method for snow depth observation and verification, but observations still could not cover the entire Arctic. The development of remote sensing has greatly promoted large-scale and long-term snow depth research. Markus and Cavalieri (1998) and Comiso et al. (2003) considered brightness temperatures measured by Special Sensor Microwave/Image (SSM/I) and Advanced Microwave Scanning Radiometer for Earth Observing System (AMSR-E)

sensors, respectively, to calculate the gradient ratio (GR) of the vertically polarized brightness temperature at 37 and 19 GHz (GRV (37/7)) and then determined the satellite-derived snow depth on sea ice. This method has often been applied to estimate the snow depth atop first-year ice (FYI) with a snow depth smaller than 50 cm. Li et al. (2017) acquired brightness temperatures from the FengYun-3B (FY-3B)/Microwave Radiation Imager (MWRI) to retrieve the snow depth based on the method of Comiso et al. (2003). Rostosky et al. (2018) successfully extended the snow depth model to determine the snow

depth atop FYI and multiyear ice (MYI) with the GRV (19/7) indicator. In contrast to snow depth models relying on linear regression analysis, Kilic et al. (2019) obtained the snow depth atop FYI and MYI via multiple linear regression analysis. The European Space Agency's Soil Moisture Ocean Salinity (SMOS) satellite provides potential data (1.4-GHz brightness temperature) for snow depth estimation (Maaß et al., 2013). In recent years, it has been verified that Ice, Cloud, and land Elevation Satellite-2 (ICESat-2) and CryoSat-2 measurements exhibit the potential to determine the snow depth on Arctic

and Antarctic sea ice (Kwok and Markus, 2017; Kowk et al., 2020; Kacimi and Kowk, 2020).

In addition to remote sensing, there are other approaches to estimate the snow depth. Warren et al. (1999) obtained Soviet drifting station data to construct the most commonly implemented large-scale snow depth climatologies from January to December, hereafter referred to as W99 climatologies. However, W99 climatologies do not suitably represent the current snow distribution on Arctic sea ice due to the gradual transition in sea ice from MYI to FYI. Shalina and Sandven (2018)

utilized data retrieved from Soviet drifting stations and Soviet airborne expeditions, producing snow depth climatologies from March to May. The reanalysis reconstruction approach is another practical method to obtain the large-scale snow depth. The NASA Eulerian Snow on Sea Ice Model (NESOSIM) was developed by Petty et al. (2018) based on atmospheric reanalysis data, determining the snow depth from August 15, 2000, to May 1, 2015, at a spatial resolution of 100 km × 100 km. Liston et al. (2018) first developed SnowModel. Thereafter, Liston et al. (2020) proposed the Lagrangian snow





evolution model (SnowModel-LG) by including physical snow processes (e.g., blowing snow sublimation and incorporation of ice dynamics), generating snow depth estimates from August 1, 1980, to July 31, 2018, at a spatial resolution of 25 km × 25 km.

      Data assimilation can effectively combine the advantages of observations and numerical models, and this method has been rapidly developed in the field of remote sensing in recent years (Dong et al., 2013). Data assimilation methods are divided

into empirical data assimilation (e.g., nudging (Bao and Errico, 1997)), statistical data assimilation (e.g., three- and four-dimensional variational assimilation (Weaver et al., 2003)) and advanced data assimilation methods (e.g., ensemble Kalman filter (Evensen et al., 1996) and particle filter (Doucet et al., 2001)). Nudging has been applied to the general circulation model by assimilating AMSR2 data, thereby improving sea ice concentration prediction (Zhao et al., 2016). As an emerging data assimilation method, the particle filter does not require the assumption of linear and Gaussian distributions. Based on an

energy-balanced snow model, Magnusson et al. (2017) assimilated snow depth estimation with a particle filter, thus generating improved snow water equivalent (SWE), snowpack runoff and soil temperature estimates. Smyth et al. (2019) applied a particle filter to assimilate the snow depth into the physical snowmelt model Snobal, and the root mean square error (RMSE) of the snow density and SWE decreased by 35% and 51%, respectively.

      Although there are many snow depth retrieval methods, the spatial coverage of satellite-derived snow depth estimates is

limited due to that of remote sensing data. The snow depth climatology product does not capture interannual variation nor does it satisfy the recent rapid change in snow depth. Although the spatial coverage of snow depth estimates obtained via simulations is large, the accuracy is relatively low. Therefore, it is urgent to develop a snow depth retrieval method that combines remote sensing and reanalysis reconstruction techniques to expand the data spatial coverage and improve the snow depth estimation accuracy. Here, an improved NESOSIM (i.e., INESOSIM) is proposed. Then, the satellite-derived snow

depth determined in our previous work is assimilated into INESOSIM with a particle filter, and the INESOSIM-PF method is developed. Finally, Arctic snow depth estimates atop sea ice from August 16 to May 15, 2012–2020, are obtained (within 10 sea areas (Fig. 1)), which provides data support for future sea ice thickness estimations.



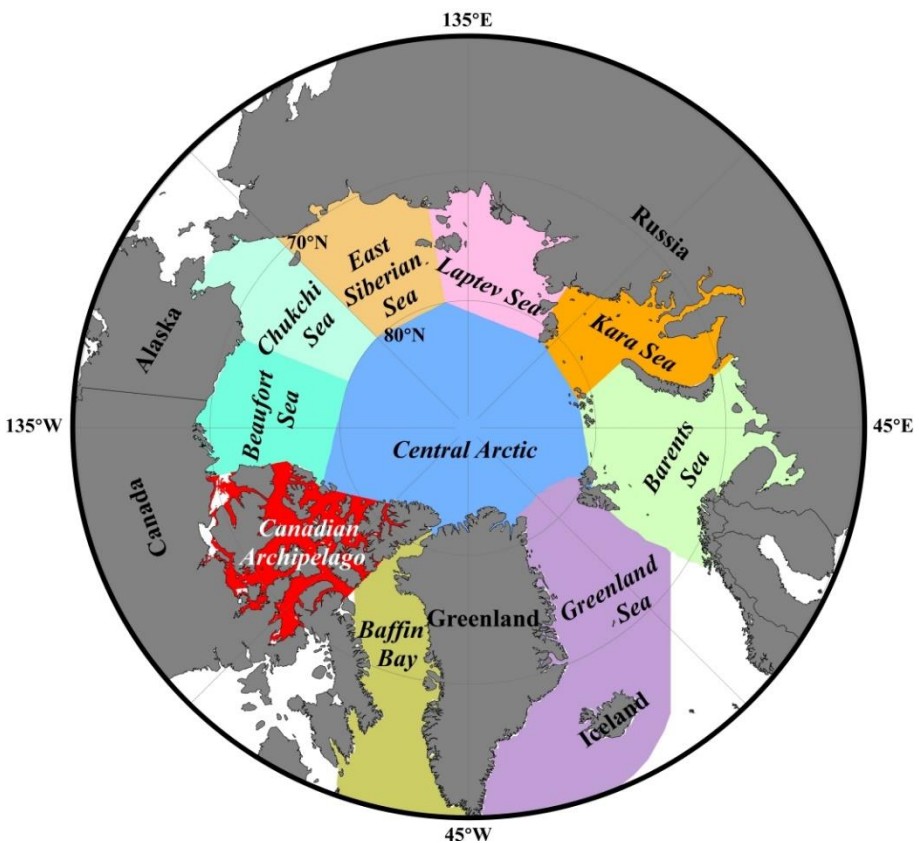

**Figure 1.** Location map of the Arctic region and its ten subregions. Note that the italics indicate the sea areas; the ten subregions are the Kara Sea, Barents Sea, Greenland Sea, Baffin Bay, Canadian Archipelago, Beaufort Sea, Chukchi Sea, East Siberian Sea, Laptev Sea, and Central Arctic.

## 2 Data

### 2.1 Reanalysis dataset

The European Centre for Medium-Range Weather Forecasts (ECMWF) provides ECMWF reanalysis 5 (ERA5) data, the latest reanalysis product of the ECMWF, for climate research. ERA5 offers a large number of atmospheric, oceanic and land variables from 1979 to the present. We choose the snowfall (sf, m of water equivalent), 10-m u-component of wind (u10, m s$^{-1}$), 10-m v-component of wind (v10, m s$^{-1}$), and 2-m temperature (2mT, K) as forcing data to run the numerical models (NESOSIM, INESOSIM and INESOSIM-PF). ERA5 data are available in Network Common Data Form (NetCDF) format with a spatial resolution of 0.25° × 0.25° and a temporal resolution of one hour. Data pertaining to the ten subregions covering the period from 2012–2020 are selected.



## 2.2 Brightness temperature and sea ice concentration

Brightness temperature (BT) and sea ice concentration (SIC) data are acquired from the National Snow and Ice Data Center (NSIDC). We select daily BT data (a spatial resolution of 25 km × 25 km) retrieved from AMSR2 measurements to obtain

the satellite-derived snow depth during the growth season (October to April) in the ten subregions from 2012 to 2020. Daily SIC data derived with the Climate Data Record (CDR) algorithm are selected as input data for NESOSIM, INESOSIM and INESOSIM-PF. The CDR algorithm combines the NASA team algorithm and NASA bootstrap algorithm. SIC data pertaining to the ten subregions covering the period from 2012–2020 are selected with a spatial resolution of 25 km × 25 km.

## 2.3 Sea ice type and sea ice drift

Sea ice type and sea ice drift data are provided by the Ocean and Sea Ice Satellite Application Facility (OSI SAF). The obtained sea ice type data exhibit a resolution of 10 km × 10 km, and the sea ice drift data exhibit a resolution of 62.5 km × 62.5 km. The sea ice type is considered to distinguish FYI and MYI. Sea ice drift is transformed to the sea ice velocity and employed to force NESOSIM, INESOSIM and INESOSIM-PF. Data pertaining to the ten subregions covering the period from 2012–2020 are selected.

## 2.4 Snow depth dataset

We select CryoSat-2 Level-4 Sea Ice Elevation, Freeboard, and Thickness Version 1 data pertaining to the ten subregions, which are provided by the NSIDC. These data provide the 30-day average snow depth on sea ice from 2010 to the present in a 25 km grid. A snow depth dataset is constructed based on a modified W99 climatology, i.e., the snow depth estimates on FYI represent half the W99 climatology, and those on MYI are equal to the W99 climatology.

## 2.5 Modeling and validation data

Operation IceBridge (OIB) data originate from the NSIDC, and these data are widely applied to evaluate satellite-derived or simulated snow depth values. OIB data from 2014 to 2017 are considered to develop the snow depth model, and OIB data from 2018 to 2019 are employed to evaluate the established snow depth model (Fig. 2(a) and 2(b)). Ice mass balance buoy (IMB) data are retrieved from the Cold Regions Research and Engineering Laboratory (CRREL)-Mass Balance Buoy

Program. This dataset is developed to monitor the sea ice volume and mass balance to better understand climate change. IMB data include the sea ice thickness, snow depth, ice surface and bottom melt. Fifteen buoys are selected from 2012 to 2018 (Fig. 2(c)), enabling the determination of the optimal parameters of the snow depth model. Data measured by snow buoys deployed during the Multidisciplinary drifting Observatory for the Study of Arctic Climate (MOSAiC) campaign are provided by the Alfred Wegener Institute (AWI), and data originating from 20 snow buoys are currently available. Among

the buoys deployed in 2019, the 2019S80 buoy only provides snow depth estimates over one month, and the estimated snow depth is smaller than 2 cm with a high uncertainty, so this buoy is excluded. Among the buoys deployed in 2020, we only


select the 2020S99 buoy, which matches our research period. Finally, the snow depth values retrieved from 12 snow buoys are considered between 2019 and 2020 to assess the accuracy of the snow depth model (Fig. 2(d)).

**Figure 2.** Spatial distribution of the (a) OIB track for modeling from 2014 to 2017, (b) the OIB track for validation from 2018 to 2019, (c) the fifteen IMB tracks for modeling from 2012 to 2018, and (d) the MOSAiC snow buoy track for validation from 2019 to 2020. Note that the legend of (c) and (d) indicates the name of the buoy.



# 3 Methods

## 3.1 Data preprocessing

We choose a 50 km × 50 km grid to run the model due to the computational efficiency and resolution of the input data. First, the SIC, snowfall, u10, v10, 2mT, and OSI SAF sea ice drift data are gridded to generate model forcing files of the same size, spatial resolution and projection. Second, the modeling and validation data are vectorized and sampled at 50-km intervals (the average value occurs within 50 km). Finally, the brightness temperature, sea ice type and NSIDC snow depth data are resampled to 50 km × 50 km.

## 3.2 Two snow depth retrieval methods

Kilic et al. (2019) developed a multilinear regression approach for snow depth estimation based on four IMB buoys, i.e., 2012G, 2012H, 2012J and 2012L. The multilinear regression relationship between the vertically polarized brightness temperatures of AMSR2 (7, 19 and 37 GHz) and the IMB-measured snow depth was established, and Eq. (1) was determined:

$$h_s=177.01+1.75 \times T_b(7V)-2.80 \times T_b(19V)+0.41 \times T_b(37V) \tag{1}$$

where $h_s$ is the snow depth atop sea ice.

Based on the snow depth data measured at Soviet stations from 1954 to 1991, Warren et al. (1999) constructed a two-dimensional quadratic function as follows:

$$h_s(x,y)=h_0+Ax+By+Cxy+Dx^2+Ey^2 \tag{2}$$

where $h_0$ is the fitted snow depth at the North Pole and $A$, $B$, $C$, $D$, and $E$ are coefficients of Eq. (2). The coefficients and $h_0$ are different in the different months. All coefficient and $h_0$ values for all 12 months are obtained from Warren et al. (1999).

## 3.3 NESOSIM

### 3.3.1 NESOSIM

Petty et al. (2018) developed a two-layer snow depth model (i.e., NESOSIM), and snow was divided into a new snow layer and an old layer. NESOSIM includes four parameterization processes.

1) Wind packing process: when the wind speed is higher than the wind threshold ($\omega$: a default value of 5 m s$^{-1}$), wind packing contributes to variations in the snow depth. Changes in the snow depth of the new snow layer are obtained with Eq. (3), and changes in the snow depth of the old snow layer are obtained with Eq. (4):

$$\Delta h_s^{wp}(t,0)= -\alpha T h_s(t,0) \tag{3}$$

$$\Delta h_s^{wp}(t,1)= (\rho_s^n/\rho_s^0)\alpha T h_s(t,0) \tag{4}$$

where $\alpha$ is the wind packing coefficient (default value: $5.8 \times 10^{-7}$ s$^{-1}$), $T$ is the daily time step (equal to 86400 s), $h_s(t,0)$ is the snow depth of the new snow layer, $h_s(t,1)$ is the snow depth of the old snow layer, and $\rho_s^n$ and $\rho_s^0$ are the new snow density





(200 kg m$^{-3}$) and old snow density (350 kg m$^{-3}$), respectively.

2) Blowing snow lost to leads: wind forcing causes any snow lost from the new snow layer to lead/open water at wind speeds higher than $\omega$. The old snow layer is wind-packed and less influenced by wind forcing.

$$\Delta h_s^{bs}(t)= -\beta TU(t)h_s(t,0)(1-SIC(t)) \tag{5}$$

where $\beta$ is the blowing snow coefficient (a default value of $2.9\times10^{-7}$ m$^{-1}$), $U$ is the wind speed, and $SIC$ is the daily sea ice concentration.

3) Ice dynamics process: the snow depth changes due to ice motion. The snow loss attributed to ice motion is divided into two terms, i.e., a divergence-convergence term and an advection term.

$$\Delta h_s^{div}(t)= -h_s(t)\cdot\nabla(u_i(t)) \tag{6}$$

$$\Delta h_s^{adv}(t)= -\nabla(h_s(t))\cdot u_i(t) \tag{7}$$

where $\Delta h_s^{div}(t)$ and $\Delta h_s^{adv}(t)$ are the variations in snow depth due to divergence-convergence and advection, respectively, and

$u_i$ is the sea ice speed.

4) Snow accumulation: snow accumulates in the snowfall process, and the snow accumulation term is calculated with the following equation:

$$\Delta h_s^{acc}(t)= S_f(t)SIC(t)/\rho_s^n \tag{8}$$

where $S_f$ is the daily snowfall within a grid.

The snow depth is calculated by adding all terms as follows:

$$h_s(t+1,\, 0)=h_s(t,\, 0) + \Delta h_s^{acc}(t)+\Delta h_s^{dyn}(t,\, 0)+\Delta h_s^{wp}(t,\, 0) + \Delta h_s^{bs}(t) \tag{9}$$

$$h_s(t+1,\, 1) = h_s(t,\, 1)+\Delta h_s^{dyn}(t,\, 1)+\Delta h_s^{wp}(t,\, 1) \tag{10}$$

### 3.3.2 Improved NESOSIM

An improved NESOSIM (INESOSIM) is proposed by including two parameterization processes: a snow melting term and a

snow lost to the atmosphere term.

With the continuous warming of the Arctic, snow melting becomes increasingly dramatic. In this study, NESOSIM starts to run in mid-August and continues to run until the following mid-May. In August and May, the snow melting process occurs in certain areas. Although snow melting is weak during these months, this process still exerts a certain impact on snow depth simulation. Therefore, we introduce the snow melting process into NESOSIM. When the 2-m temperature ($T_{air}$)

is higher than 0 °C, we consider that there occurs a snow melting process on sea ice.

$$\Delta h_s^{melt}(t)= -T_{air}(t)T\tau\rho_w/\rho_s^n \tag{11}$$

where $\tau$ is the degree-day factor and $\rho_w$ is the water density. We set $\tau$ to $6.3\times10^{-8}$ m °C$^{-1}$ s$^{-1}$ (Kuchment and Gelfan, 1996), which is determined via the degree-day method.

In addition to wind packing and blowing snow loss to leads caused by wind, wind transports snow into the atmosphere,

resulting in a reduction in snow depth. In 2020, Petty (2020) updated the model and proposed that the snow lost to the


atmosphere process should be considered. However, the updated algorithm has not been debugged, and further tests are needed to determine the corresponding parameters. Similar to the blowing snow lost to leads and wind packing processes, snow is lost to the atmosphere when $U$ exceeds 5 m s$^{-1}$. The atmospheric loss term is determined by the blowing snow coefficient, atmospheric loss coefficient ($\gamma$), wind speed and depth of the new snow layer. The equation is as follows:

$\quad \Delta h_s^{atm}(t) = -\beta \gamma U T h_s(t, 0)$ (12)

### 3.4 Particle filter assimilation

In recent decades, the particle filter has become a popular data assimilation approach. The core of the particle filter is Monte Carlo simulation and importance resampling. Here, we provide a simple description of the particle filter, and more details are found in Arulampalam et al. (2002), Smyth et al. (2019) and Magnusson et al. (2017). The particle filter contains four steps:

$\quad$ a prediction step, update step, resampling step and output step.

1) Prediction step

A. The initial state variable is set, i.e., $x_k$. Random noise with an arbitrary distribution is provided to disturb $x_k$, and the n-dimensional initial state $x_k^i$ is obtained at time step k.

B. The weight of each particle is set to 1/N, and N is the number of particles

$\quad$ C. State and measurement prediction:

$x_{k+1}^i = f\left(x_k^i, \theta_k^i, u_k\right) + v_k$ (13)

$z_{k+1}^i = h(x_{k+1}^i) + n_{k+1}$ (14)

where $f$ and $h$ are the state function and measurement function, respectively, $x$ and $z$ are the state vector and measurement vector, respectively, $\theta$ is a model parameter, $u$ is the model input, and $v$ and $n$ are the process noise and measurement noise,

$\quad$ respectively.

2) Update step

A. The weight of each particle is calculated:

$w_{k+1}^i = w_k^i p(z_{k+1} | x_{k+1}^i)$ (15)

$p(z_{k+1} | x_{k+1}^i) = \dfrac{1}{\sqrt{(2\pi)^N |C_v|}} e^{[-0.5(z_{k+1}-x_{k+1}^i)(z_{k+1}-x_{k+1}^i)/C_v]}$ (16)

$\quad$ where $p$ is the likelihood function and $C_v$ is the measurement error covariance.

B. The weights are normalized, namely, the sum of all weights equals 1.

3) Resampling step

According to the weight, any particles with a low weight are discarded, and the particles with a high weight are duplicated. After importance resampling, the total number of particles remains the same. Then, the weight is reset to 1/N.

$\quad$ 4) Output step

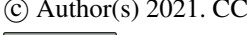



The mean of the ensemble (Dong et al., 2015) is selected to define the best estimates of the state vector. Then, the best estimates are output.

## 4 Results

### 4.1 INESOSIM-PF modeling

**4.1.1 Determination of NESOSIM**

Petty et al. (2018) performed sensitivity analysis of three model parameters ($\alpha$, $\beta$, and $\omega$). The results indicated that NESOSIM was sensitive to $\alpha$. At an $\alpha$ value of $5.8\times10^{-7}$ s$^{-1}$, the simulated snow depth was the most consistent with the obtained data from Soviet drifting stations. When $\alpha$ equaled $11.6\times10^{-7}$ s$^{-1}$, the snow density was greatly influenced. When the blowing snow coefficient was set to twice the default value (i.e., $\beta$ was $5.8\times10^{-7}$ m$^{-1}$), negligible effects were observed on the

simulation results. Therefore, the model was insensitive to $\beta$. When the wind threshold was 10 m s$^{-1}$, the difference between the snow depth and drifting station data notably increased. Therefore, when we run NESOSIM, $\alpha$ and $\omega$ are set to default values. Then, adopting OIB data from 2014 to 2017, we perform experiments to select an appropriate $\beta$ value.

When the default value is applied, the bias between the simulated snow depth and OIB-measured snow depth is 10.79 cm (Fig. 3). When a value of $5.8\times10^{-7}$ m$^{-1}$ is applied, the deviation from the OIB data slightly decreases (Fig. 3). When a value

of $11.6\times10^{-7}$ m$^{-1}$ is applied, the deviation is reduced to 7.74 cm, which greatly improves the accuracy of the simulated snow depth (Fig. 3). NESOSIM generally overestimates the snow depth in grid cells, and an increased $\beta$ value can alleviate this snow depth overestimation but can increase the deviation in certain underestimated grid cells (Fig. 3). Most grid cells exhibit a small decrease in snow depth deviation, and a small number of grid cells exhibits a large decrease in snow depth deviation (Fig. 3). This indicates that the selection of these three values ($2.9\times10^{-7}$ m$^{-1}$, $5.8\times10^{-7}$ m$^{-1}$ and $11.6\times10^{-7}$ m$^{-1}$) exerts less

impact on the simulation results in most regions. We finally choose a value of $11.6\times10^{-7}$ m$^{-1}$ to obtain the most accurate snow depth estimates among the three simulated snow depth vectors.



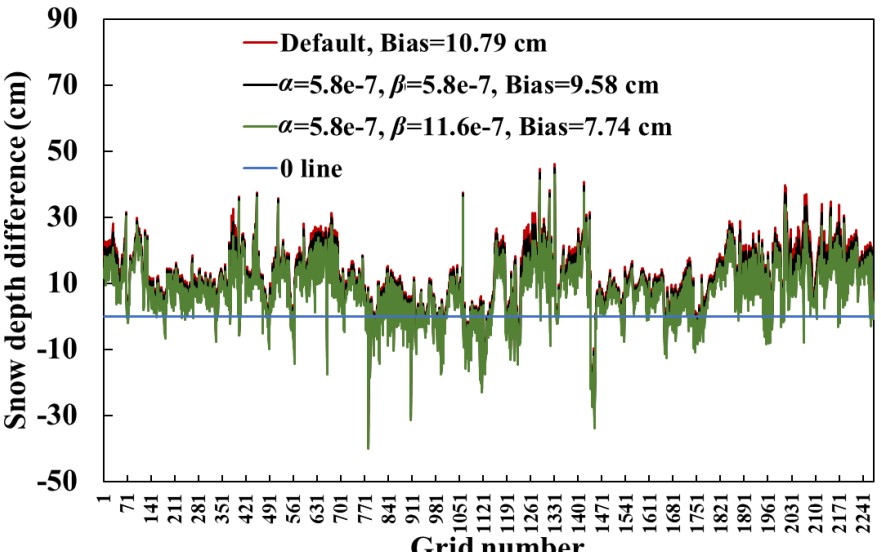

**Figure 3.** Snow depth difference (the simulated snow depth minus the OIB-measured snow depth) in the different matching grids from 2014–2017.

### 4.1.2 Determination of INESOSIM

First, we incorporate a melting term into the four basic parameterization processes (NESOSIM) to verify whether the added melting term improves the model accuracy. Compared to NESOSIM, the snow depth in most matching grids decreases, and the snow depth in a few matching grids remains unchanged (Fig. 4(a)). In certain grids, due to the stronger melting process, there are greater reductions in snow depth. The overall accuracy is improved to a certain extent (RMSE: decreases 4%; bias: decreases 8%; MAE: decreases 4%; P: decreases 1%). Under global warming, the effect of the snow melting process will become increasingly obvious. Therefore, considering the melting term is necessary and helpful to improve the model accuracy.

Next, we add the atmospheric loss term. The amount of snow lost is determined by the atmospheric loss coefficient $\gamma$, and experiments are carried out to determine $\gamma$. The snow depth bias without the atmospheric loss term is 7.15 cm (Fig. 4(b)), which greatly overestimates the snow depth. Then, $\gamma$ values of 0.0125, 0.015, 0.020 and 0.025 are tested. The results suggest that including the atmospheric loss term greatly reduces the bias between the simulated snow depth and OIB-measured snow depth, namely, with increasing coefficient, the bias decreases. When we set $\gamma$ equal to 0.025, the bias is only 0.58 cm (Fig. 4(b)). According to the accuracy, three options (0.015, 0.020 and 0.025) are chosen to determine the corresponding distribution as well as the OIB snow depth distribution (Fig. 4(c)). When $\gamma$ is equal to 0.015, snow depths greater than 48 cm cannot be simulated by INESOSIM, but when $\gamma$ is equal to 0.020, INESOSIM is ineffective at snow depths greater than 45 cm, whereas when $\gamma$ is equal to 0.025, INESOSIM is ineffective at snow depths greater than 43 cm. With increasing atmospheric loss coefficient, the retrieval ability of INESOSIM for thick snowpacks is weakened.





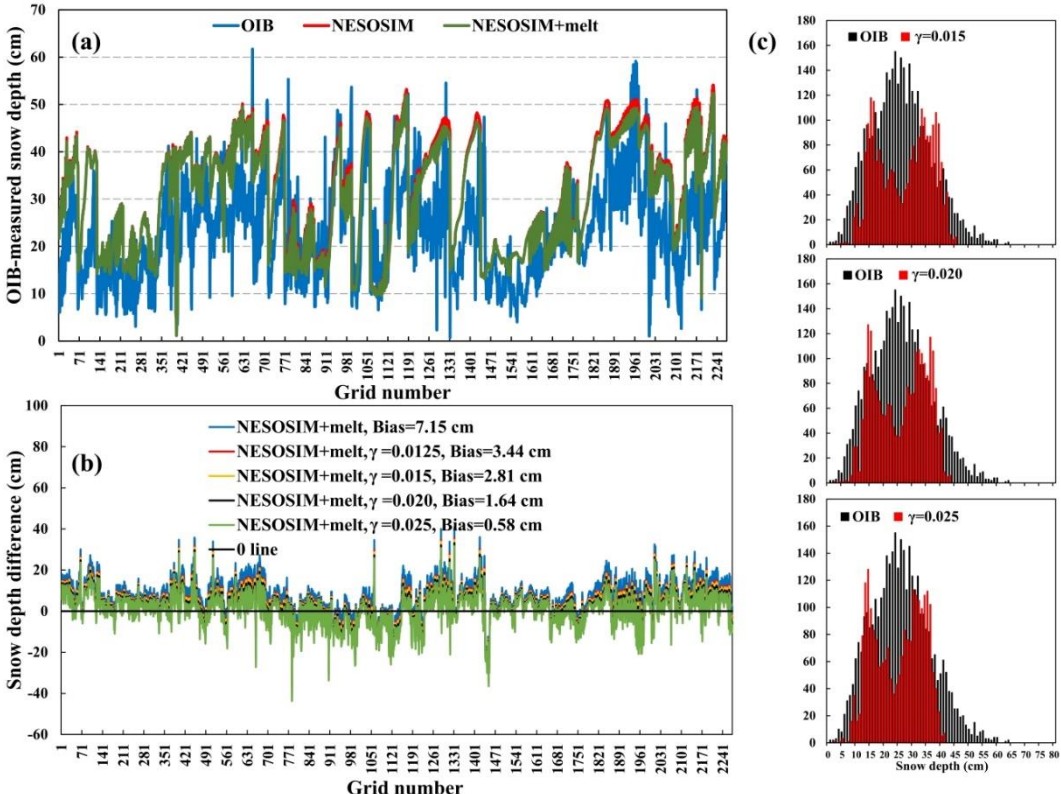

**Figure 4.** (a) Snow depth retrieved from OIB, NESOSIM and NESOSIM with a melting process in the different matching grids; (b) snow
275   depth difference in the different matching grids considering various atmospheric loss coefficient values; (c) distribution of the OIB-
measured snow depth in black versus that of the simulated snow depth considering different $\gamma$ values in red. The data cover the period from
2014 to 2017.

To determine the optimal $\gamma$ value among the three options of 0.015, 0.020 and 0.025, we compare the three snow depth
vectors to the IMB-measured snow depth and choose the optimal atmospheric loss coefficient that yields the most accurate
snow depth based on the IMB-measured snow depth. For convenience, the snow depths obtained by adopting 0.015, 0.020
and 0.025 are denoted as the F1 snow depth, F2 snow depth and F3 snow depth, respectively. The biases between the F1, F2
and F3 snow depths and the IMB-measured snow depth are much smaller than 0, indicating great snow depth
underestimation. Among the F1, F2 and F3 snow depths, the error in the F1 snow depth is the smallest. The RMSE of the F1
snow depth decreases by 3.06%, the bias decreases from -7.97 cm to -6.67 cm (improves by 16.18%), the MAE decreases by
3.88% and the correlation increases by 5.18% over the F3 snow depth (Table 1). Compared to the F3 snow depth, the RMSE
of the F2 snow depth decreases by 1.53%, the bias improves by 7.73%, the MAE decreases by 1.96%, and the correlation
increases by 2.60% (Table 1). In summary, we select 0.015 to obtain the atmospheric loss term. Hence, all the INESOSIM
parameters are determined.






**Table 1.** Accuracy of INESOSIM with different atmospheric loss coefficient values ($\gamma$) based on the IMB-measured snow depth (number of matching points (N), RMSE (cm), bias (cm), MAE (cm) and r).

| $\gamma$ | 0.015 | 0.020 | 0.025 |
|---|---|---|---|
| N | 443 | 443 | 443 |
| RMSE (cm) | 16.58 (3.06%) | 16.85 (1.53%) | 17.11 |
| Bias (cm) | -6.67 (16.18%) | -7.36 (7.73%) | -7.97 |
| MAE (cm) | 11.06 (3.88%) | 11.28 (1.96%) | 11.51 |
| r | 0.12 (5.18%) | 0.12 (2.60%) | 0.11 |

### 4.1.3 Determination of INESOSIM-PF

We use satellite-derived snow depth values based on regression analysis (RA) and 5 variables (GR$_V$ (37/19), GR$_H$ (37/19),
GR$_V$ (37/7), GR$_V$ (19/10) and GR$_V$ (19/7)) of the long short-term memory (i.e., RA-5VLSTM snow depth) (Li and Ke, 2021), which was established in our previous work, to assimilate the INESOSIM snow depth. Before data assimilation, the following postprocessing step was performed. The satellite-derived snow depth with a 25 km × 25 km grid resolution is resampled to a 50 km × 50 km grid resolution. According to the boundary of the study area, the satellite-derived snow depth and the simulated snow depth are clipped to ensure data assimilation.

First, after matching the OIB-measured snow depth with the satellite-derived snow depth, the initial state value is set to 28.51 cm, and the actual measured value is set to 15.64 cm. The number of particles is set to 1000. The more particles, the closer the assimilation result is to the real value. Smyth et al. (2019) set the number of particles to 100, and effective results were obtained. Therefore, we choose 1000 to ensure a sufficient number of particles and obtain reliable results. Both the process noise variance and observation noise variance are set to 5 cm. Smyth et al. (2019) tested observation noise variance
values of 2 cm and 10 cm and concluded that the assimilation results were insensitive to the observation noise variance.

### 4.2 Accuracy evaluation

To evaluate the accuracy and superiority of the assimilation results (INESOSIM-PF), we compare the obtained NESOSIM, INESOSIM, RA-5VLSTM, and INESOSIM-PF snow depth estimates to independent OIB measurements from 2018 to 2019 and MOSAiC measurements from 2019 to 2020. According to the OIB data, INESOSIM greatly
improves the simulated snow depth over NESOSIM. The RMSE decreased 9.38%, the MAE decreased 8.19%, and the previously positive bias developed into a negative bias (from 2.81 cm to -1.75 cm) (Table 2). In regard to the RA-5VLSTM snow depth retrievals, the RMSE reached 6.24 cm, and the correlation coefficient was 0.76, indicating that the accuracy is much higher than that of the NESOSIM and INESOSIM snow depths. Compared to INESOSIM, the accuracy of the INESOSIM-PF snow depth has been greatly improved, namely, the RMSE decreases by 9.48%, the MAE decreases
by 11.50%, the bias changes from -1.75 cm to -1.40 cm (improves by 20.04%), and the correlation coefficient increases by 13.64% (Table 2). Although INESOSIM-PF generates a slightly lower accuracy than that obtained with the RA-5VLSTM method, it generates more matching points with the OIB data (Table 2).





According to the MOSAiC snow buoys, the snow depth estimates obtained with the four methods are generally higher than the MOSAiC-measured snow depths. NESOSIM attains the highest RMSE value of 13.93 cm among the four snow

depth models (Table 2). Compared to the snow depth estimates retrieved from NESOSIM, the snow depth estimates obtained with INESOSIM-PF are greatly improved (RMSE: decreases by 33.76%; bias: decreases by 37.73%; MAE: decreases by 30.16%; r: increases by 18.55%) (Table 2). Compared to RA-5VLSTM, the accuracy of INESOSIM-PF is slightly higher (RMSE: decreases by 3.31%; bias: decreases by 2.25%; MAE: decreases by 3.09%; r: increases by 1.34%) (Table 2).

**Table 2.** Accuracy evaluation of the NESOSIM, INESOSIM, RA-5VLSTM and INESOSIM-PF methods through the number of matching points (N), RMSE (cm), bias (cm), MAE (cm) and r based on the OIB-measured snow depth from 2018 to 2019 and the MOSAiC-measured snow depth from 2019 to 2020.

| | OIB | | | |
|---|---|---|---|---|
| | NESOSIM | INESOSIM | RA-5VLSTM | INESOSIM-PF |
| N | 528 | 528 | 428 | 528 |
| RMSE (cm) | 8.20 | 7.43 | 6.24 | 6.73 |
| Bias (cm) | 2.81 | -1.75 | -1.35 | -1.40 |
| MAE (cm) | 6.25 | 5.74 | 5.02 | 5.08 |
| r | 0.64 | 0.63 | 0.76 | 0.71 |
| | MOSAiC | | | |
| N | 71 | 71 | 64 | 71 |
| RMSE (cm) | 13.93 | 9.95 | 9.54 | 9.23 |
| Bias (cm) | 11.96 | 7.53 | 7.62 | 7.44 |
| MAE (cm) | 12.29 | 8.52 | 8.86 | 8.59 |
| r | 0.27 | 0.27 | 0.29 | 0.32 |

Because the number of matching grids of the RA-5VLSTM-determined snow depth is the smallest, we unify the matching

grids of all snow depth estimates to be consistent with those of the RA-5VLSTM snow depth to obtain comparable results of the snow depth distribution. The results indicate that NESOSIM is ineffective at snow depths smaller than 15 cm, and the deviation between the NESOSIM-estimated snow depth values and OIB data ranges from -10 cm to 20 cm, mostly located on the right side of the 0 line, suggesting snow depth overestimation (Fig. 5(a)). The INESOSIM snow depth estimates vary between 10 cm and 45 cm, and the model is inapplicable for thick ice with snow depths greater than 43 cm and thin ice with

snow depths less than 10 cm. The deviation between the INESOSIM-estimated snow depth values and OIB data mainly varies between -15 cm and 10 cm, and more deviation values are located on the left side of the 0 line, suggesting snow depth underestimation (Fig. 5(b)). Compared to the distribution of the NESOSIM snow depth difference, the distribution of the INESOSIM snow depth difference is more concentrated. The RA-5VLSTM snow depth values vary between 5 cm and 50 cm, which performs well for thick snowpacks. The distribution of snow depth values greater than 30 cm is basically

consistent with that of the OIB measurements. The deviation between the RA-5VLSTM snow depth estimates and OIB measurements ranges from -10 cm to 10 cm and evenly distributed on both sides of the 0 line (Fig. 5(c)). The INESOSIM-





PF-estimated snow depth is distributed between 10 cm and 45 cm, and its distribution agrees well with the distribution of the RA-5VLSTM snow depth (Fig. 5(c) and (d)), indicating that the INESOSIM-PF snow depth estimates are close to the satellite-derived snow depth values and that the simulation ability of thick snow packs is enhanced after data assimilation.

Furthermore, the similarity between the simulated snow depth distribution and OIB snow depth distribution increases (Fig. 5 (d)). The INESOSIM-PF snow depth deviation varies between -10 cm and 10 cm, and the INESOSIM-PF snow depth inherits the characteristics of a high accuracy of the satellite-derived snow depth.

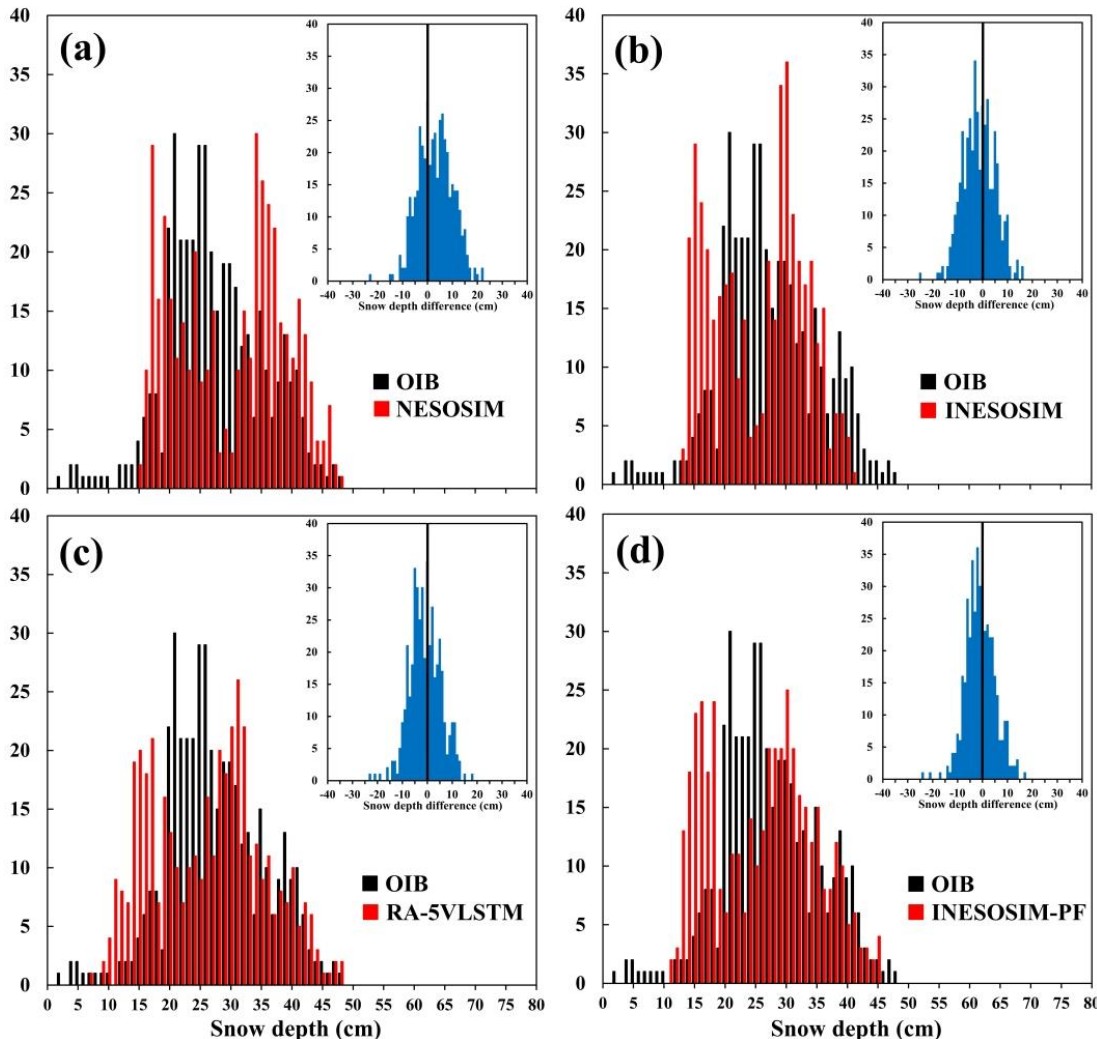

**Figure 5.** (a) Distribution of the OIB-measured snow depth in black versus the estimated snow depth using the different algorithms in red;
(b) deviation between the estimated snow depth using the different algorithms and the OIB-measured snow depth from 2018 to 2019.





## 4.3 Comparison of the snow depth determined with the different methods

We define the snow depth obtained from Kilic et al. (2019) as the Kilic19 snow depth and that acquired from the NSIDC as the NSIDC snow depth for convenience. We obtain the error in the Kilic19 snow depth based on the OIB-measured snow depth from 2018 to 2019. The results indicate that the RMSE of the Kilic19 snow depth is 11.02 cm, which is approximately

1.6 times that of the INESOSIM-PF snow depth (Tables 2 and 3, respectively). According to the MOSAiC snow buoys, the Kilic19 model generates a slightly smaller RMSE and a greatly lower correlation coefficient than does INESOSIM-PF (Table 3). The RMSE of the NSIDC model reaches 13.33 cm, and the correlation coefficient is the highest of 0.34 among all six methods (Tables 2 and 3, respectively). The above results indicate that the Kilic19 method performs well in snow buoy distribution areas, while the INESOSIM-PF method performs well in both snow buoy and OIB distribution areas

(e.g., the Beaufort Sea and Chukchi Sea).

**Table 3.** Accuracy evaluation of the Kilic19 and NSIDC methods through the number of matching points (N), RMSE (cm), bias (cm), MAE (cm) and r based on the OIB-measured snow depth from 2018 to 2019 and the MOSAiC-measured snow depth from 2019 to 2020.

|  | OIB | | MOSAiC | |
|---|---|---|---|---|
|  | Kilic19 | NSIDC | Kili19 | NSIDC |
| N | 503 | / | 65 | 71 |
| RMSE (cm) | 11.02 | / | 8.91 | 13.33 |
| Bias (cm) | 1.92 | / | 6.09 | 11.14 |
| MAE (cm) | 9.17 | / | 7.99 | 11.45 |
| r | 0.72 | / | 0.01 | 0.34 |

The Kilic19, NSIDC and INESOSIM-PF snow depth estimates during the winter 2019/2020 period are compared. The results indicate that the Kilic19 snow depth coverage is the lowest, the snow depth estimates over MYI are the largest among the three snow depth datasets, and there are large areas with snow depths greater than 44 cm (Fig. 6(a)). The INESOSIM-PF snow depth coverage is the highest, and the snow depth in small regions in the MYI areas is larger than 44 cm (Fig. 6(a)). The snow depth on MYI provided by the NSIDC is the smallest, and the maximum value is smaller than 44 cm (Fig. 6(a)).

The spatial distribution of the snow depth estimated with the three methods is similar, with the largest estimates in the Central Arctic and small estimates in the Chukchi Sea, the East Siberian Sea and the Laptev Sea (Fig. 6(a)). The large difference between the Kilic19 and INESOSIM-PF snow depth estimates is mainly distributed in the Central Arctic, which is covered by MYI. In MYI areas, the difference in snow depth between large sea areas near Greenland and the Canadian Archipelago is larger than 10 cm, and the difference in snow depth between other areas varies between -5 cm and 5 cm (Fig.

6(b)). The difference between the NSIDC and INESOSIM-PF snow depth estimates is much smaller than that between the Kilic19 and INESOSIM-PF snow depth estimates. Only a small region in the nearshore area exhibits a snow depth difference larger than 10 cm (Fig. 6(b)). Except for a part of the MYI area, the NSIDC snow depth estimates are greater than the INESOSIM-PF snow depth estimates, and the deviation varies between 0 cm and 5 cm (Fig. 6(b)).





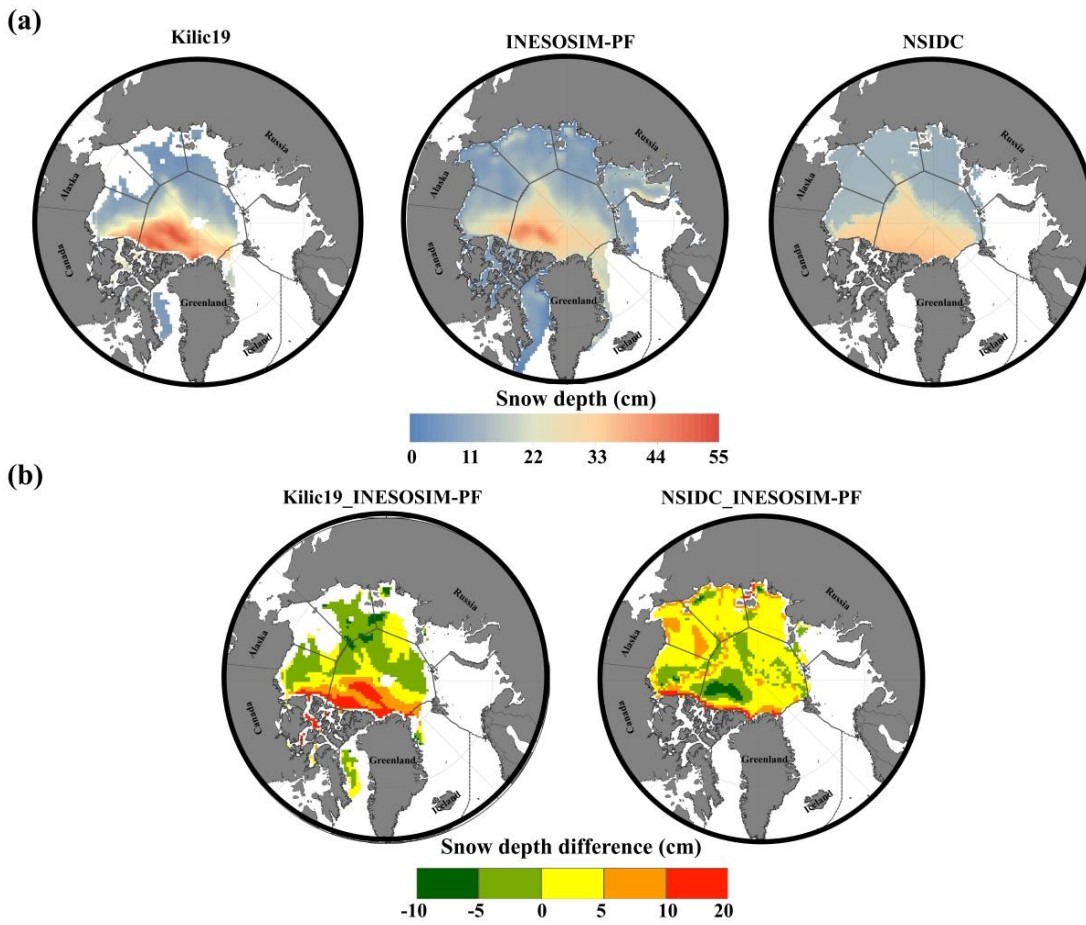

**Figure 6.** (a) Spatial distribution of the snow depth based on the different algorithms (Kilic19, INESOSIM-PF and NSIDC) from January to March 2020 (i.e., winter 2019/2020); (b) deviation between the estimated snow depth using two algorithms (Kilic19 and NSIDC) and the INESOSIM-PF snow depth during the winter 2019/2020 period.

Since the spatial coverage of the Kilic19 snow depth dataset is the lowest, we employ the intersection area of the monthly average Kilic19 snow depth to determine the common region (CR) and obtain snow depth estimates from Kilic19, INESOSIM-PF and NSIDC (Fig. 7(a) and 7(b)). Changes in the snow depth based on these three datasets in the CR during the cold season are compared. The results indicate that the variations in the Kilic19 snow depth estimates are basically consistent with those in the INESOSIM-PF snow depth estimates, but the Kilic19 snow depth estimates are significantly larger than the INESOSIM-PF snow depth estimates. The former estimates vary between 30 cm and 70 cm, and the latter estimates vary between 20 cm and 50 cm (Fig. 7(b)). There are differences in the monthly snow depth variation in the different years, i.e., from October 2015 to April 2016, the snow depth increased, while the monthly average snow depth first increased and then decreased during the frozen season from 2016–2020 (Fig. 7(b)). There is no interannual variation in the NSIDC snow depth. The monthly average snow depth during the cold season first rapidly increases, then decreases and finally increases. The variations in the snow depth range from 30 cm to 40 cm (Fig. 7(b)). The autumn Kilic19 and





INESOSIM-PF snow depths generally exhibit a downward trend. The former varies between 40 cm and 60 cm, and the latter
varies between 30 cm and 40 cm (Fig. 7(c)). The winter Kilic19 and INESOSIM-PF snow depth variations are basically
consistent (Fig. 7(d)). The former varies between 40 cm and 60 cm, and the latter varies between 30 cm and 45 cm (Fig.
7(d)). Generally, the Kilic19 model yields larger snow depth estimates in the CR. The INESOSIM-PF snow depth variation
is consistent with the Kilic19 snow depth variation, and the snow depth value is closer to the NSIDC snow depth, indicating
that the INESOSIM-PF snow depth is reliable.

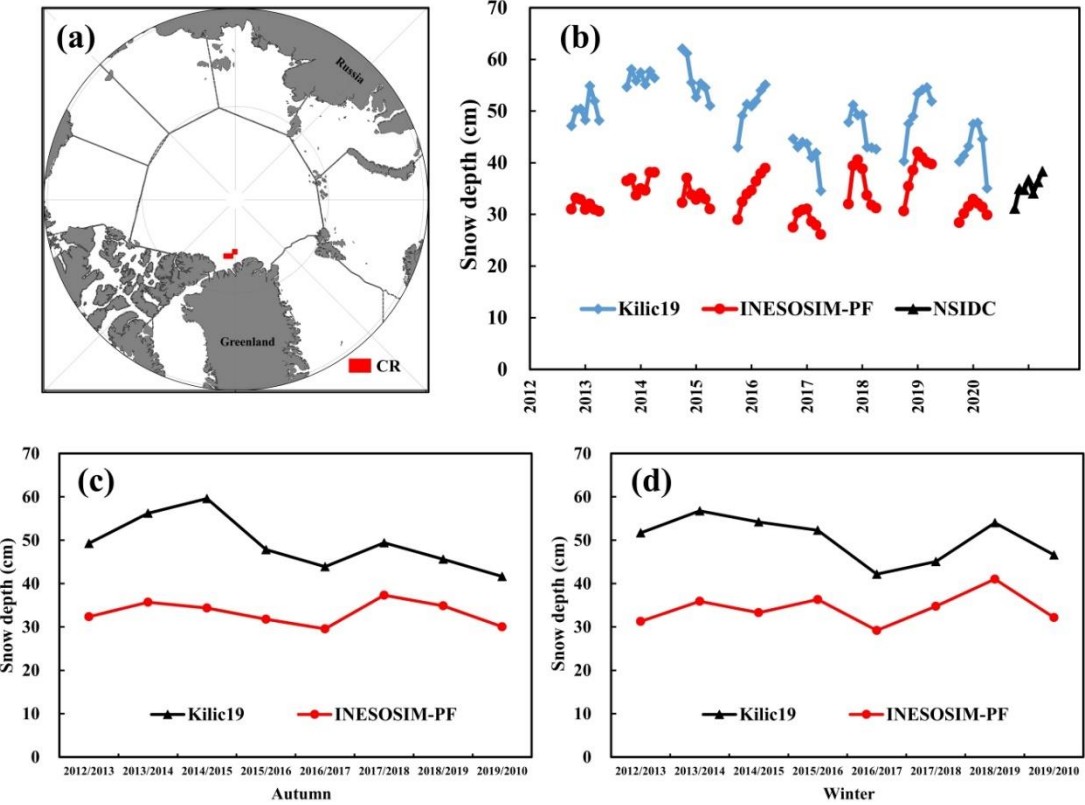

**Figure 7.** (a) Spatial distribution in the common region (CR) in the Central Arctic; (b) time series of the monthly snow depth based on the
different algorithms (Kilic19, INESOSIM-PF, and NSIDC) in the CR from 2012 to 2020; time series of (c) the autumn and (d) winter
snow depths based on the different algorithms (Kilic19, INESOSIM-PF) in the CR from 2012 to 2020.

## 5 Discussion

### 5.1 Particle number sensitivity

The sensitivity of INESOSIM-PF to the number of particles in the Arctic from 2012–2020 is examined. We show the results
for different numbers of particles, i.e., 50, 250, 500, 750, 1000 and 1250 (1000 is the default value). For convenience, the
snow depths obtained with 1000, 50, 250, 500, 750, and 1250 particles are defined as the default snow depth, P50 snow



depth, P250 snow depth, P500 snow depth, P750 snow depth and P1250 snow depth, respectively. The default snow depth
begins to increase rapidly in mid-August, which continues until October (Fig. 8(a)). In October, the increase rate of the snow
depth decreases, and the snow depth begins to decline in late October. This change can also be observed among the results of
Petty et al. (2018) using the snowfall products of JAR-55 and MEDIAN-SF to run NESOSIM. However, this change is more
notable in our results. Then, the snow depth slowly increases until May due to the continuous accumulation of snow and less
snow melting during the cold season. The interannual variability in the snow depth from August to September is low,
basically between -1 cm and 1 cm (Fig. 8(a)). The annual variability in the snow depth from October to May ranges from -
2.01 cm to 2.01 cm.

In the Arctic, the absolute differences between the monthly P50 snow depth and default snow depth are large from
October to December, reaching a maximum value of 4.10 cm in November (Fig. 8(b)). The differences between the monthly
P250 snow depth, P500 snow depth, P750 snow depth and default snow depth are larger than 0. The closer the number of
applied particles is to the default value, the smaller the snow depth difference. The differences between the monthly P1250
snow depth and default snow depth are smaller than 0. The largest snow depth difference is smaller than 0.01 cm (Fig. 8(b)).
All the absolute values of the five average monthly snow depth difference vectors increase rapidly, a maximum value is
reached from November-January, and the value subsequently decreases (Fig. 8(b)).

The Central Arctic is an area largely covered with thick snow. Through the above sensitivity analysis in this area, we can
elucidate the influence of the particle number on the area with a large snow depth. When the particle number is small (i.e.,
50), the choice of the particle number imposes a dramatic effect on the snow depth in thick-snow areas, with the largest
deviation of 10.28 cm in November (Fig. 8(c)). When the particle number is larger than 250, the choice of the particle
number yields no effect on the snow depth in thick-snow areas, and the absolute difference between the four snow depth
vectors and the default snow depth is smaller than 0.014 cm (Fig. 8(c)). After January, the absolute value of the five average
monthly snow depth differences remains small (0.002 cm) (Fig. 8(c)).

The Chukchi Sea is covered with thin FYI, and the snow depth is small. The influence of the particle number on areas
with small snow depths can be revealed by analyzing the Chukchi Sea. The absolute differences between the monthly P50
snow depth and default snow depth are the largest in October, with an absolute value of 1.22 cm (Fig. 8(d)). The snow depth
difference based on the other four numbers of particles (250, 500, 750, and 1250) is smaller than 0.10 cm (Fig. 8(d)). This
verifies that when the number of particles is large, the choice of the particle number imposes little effect on snow depth
estimation.





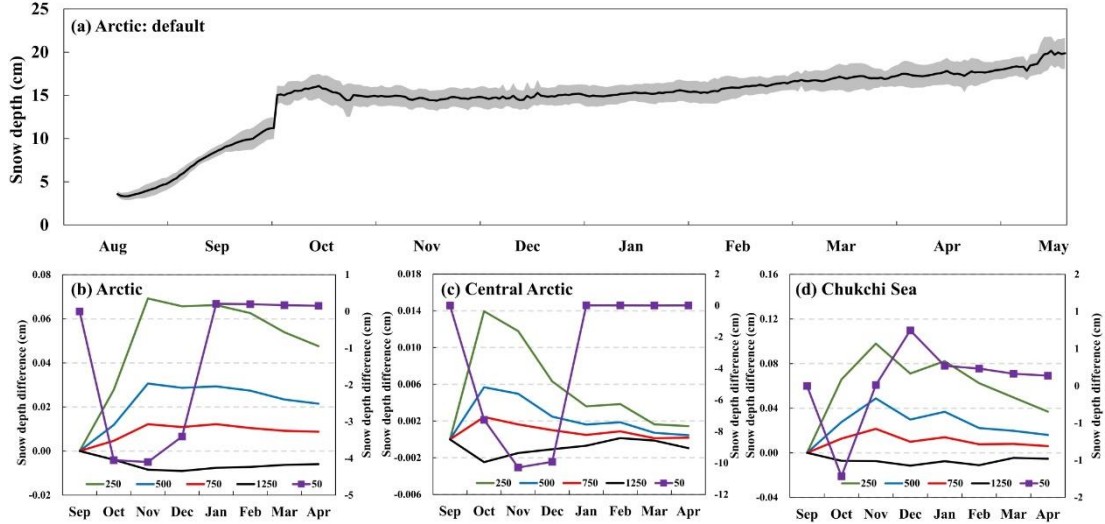

**Figure 8.** (a) Variations in the daily average default snow depth in the Arctic from 2012 to 2020, and the shaded areas represent the interannual variability using 1 standard deviation; variations in the monthly average snow depth difference (snow depth estimates based on the different particle numbers minus the default snow depth) in the (b) Arctic, (c) Central Arctic and (d) Chukchi Sea. Note that the secondary axes of (b), (c) and (d) indicate the difference between the P50 snow depth and the default snow depth.

## 5.2 Uncertainties, advantages and disadvantages of INESOSIM-PF

We adopt the Monte Carlo method (Braakmann-Folgmann and Donlon, 2019) to determine the INESOSIM-PF snow depth uncertainty. The satellite-derived snow depth contains an uncertainty of 1 cm, and the NESOSIM snow depth uncertainty reaches 5 cm (Petty et al., 2020). For each input data source, we randomly generate 50 samples, considered as an ensemble, to run the model. Then, the model generates a group of snow depth estimates for each ensemble. The standard deviation in the various snow depth estimate groups is defined as the uncertainty in each snow depth group. Fig. 9 shows the uncertainty in each estimated snow depth group. An uncertainty lower than 1 cm accounts for 80% of all uncertainties. The maximum uncertainty reaches 3.10 cm, and the minimum value is 0.29 cm. After averaging all group uncertainty values, the final uncertainty is obtained, i.e., 0.74 cm, indicating that the model is robust. The application of a particle filter greatly reduces the uncertainty caused by the uncertainty in the input parameters. However, the Monte Carlo method cannot assess the uncertainty caused by the modeling process, which should be further evaluated in the future.





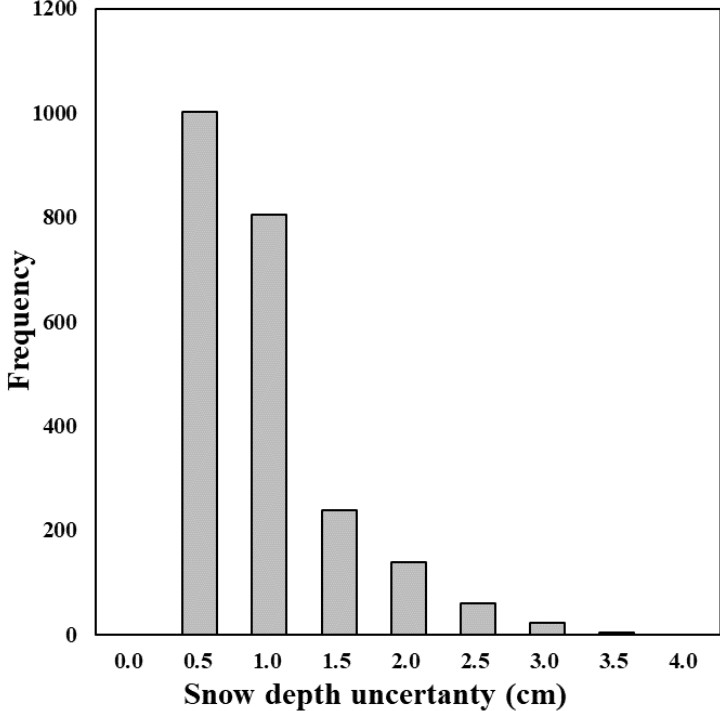

**Figure 9.** Distributions of the snow depth uncertainty across all the OIB-measured snow depths from 2014 to 2019.

INESOSIM-PF effectively combines the advantages of snow depth estimation from reanalysis reconstructions and remote sensing and notably reduces the impact of the input data uncertainty. The estimated INESOSIM-PF snow depth exhibits a large spatial coverage and provides a daily snow depth dataset from August 16 to May 15, 2012–2020. This dataset also inherits the high accuracy of the satellite-derived snow depth. The process of INESOSIM-PF is simple and easy to understand. The model achieves a high operation speed and is suitable for snow depth estimation atop FYI and MYI.

However, there are certain limitations in INESOSIM-PF. First, INESOSIM-PF depends on assimilation data to a great extent. Based on OIB-measured snow depth values from 2018 to 2019, the accuracy of the estimated snow depth when applying the Kilic19 snow depth as assimilation data (RMSE: 9.77 cm; bias: 1.22 cm; MAE: 8.07 cm; r: 0.69) is lower than the accuracy of the estimated snow depth when applying the RA-5VLSTM snow depth as assimilation data (Table 2). The closer the assimilation data are to the real values, the higher the snow depth accuracy of INESOSIM-PF. Therefore, it is very

important to choose high-quality assimilation data. Second, NESOSIM is sensitive to snowfall data retrieved from different reanalysis datasets, sea ice motion and sea ice concentration data (Petty et al., 2018), and INESOSIM-PF does not avoid this problem. Therefore, the choice of forcing data exerts a great impact on snow depth estimation and cannot be ignored. Third, due to severe snow and sea ice melting, this method does not estimate the snow depth from mid-May to mid-August.





## 6 Conclusions

To better understand variations in the snow depth and sea ice, we develop INESOSIM-PF based on reanalysis reconstruction and data assimilation methods. First, the coefficients of NESOSIM are determined by considering the OIB-measured snow depth. Then, we include a snow melting term and a snow lost to the atmosphere term to establish INESOSIM. The atmospheric loss coefficient of INESOSIM is determined based on OIB-measured and IMB-measured snow depth values. Next, the satellite-derived snow depth (RA-5VLSTM snow depth) is assimilated via a particle filter, and the final

INESOSIM-PF model is established to yield snow depth estimates from August 16 to May 15, 2012–2020. This greatly solves the problem that W99 climatologies do not suitably reflect the current changes in snow depth.

Based on OIB-measured snow depth values from 2018 to 2019, the INESOSIM-estimated snow depth is greatly improved over the NESOSIM-estimated snow depth. The INESOSIM-PF-estimated snow depth is further improved over the INESOSIM-estimated snow depth, namely, the RMSE decreases by 9.48%, and the correlation coefficient increases by

13.64%. Compared to the RA-5VLSTM snow depth employed for assimilation, the matching points of the INESOSIM-PF-estimated snow depth and OIB-measured snow depth increase by 23.36%, and the spatial coverage is highly improved. Based on MOSAiC-measured snow depth values, the INESOSIM-PF-estimated snow depth is more accurate than the RA-5VLSTM and INESOSIM snow depth estimates. The INESOSIM-PF-estimated snow depth is insensitive to the selection of the particle number when the particle number is larger than 250. The model is robust, uncertainties are concentrated within 1

cm, and the average uncertainty is 0.74 cm.

The spatial distribution of the snow depth retrieved from the INESOSIM-PF, Kilic19 and NSIDC methods is consistent. Except for the snow depth difference between the INESOSIM-PF and NSIDC approaches in the nearshore area smaller than 10 cm, the snow depth difference is smaller than 5 cm in the other sea areas. The Kilic19 snow depth is larger than the INESOSIM-PF-estimated snow depth in MYI regions. The monthly and seasonal (referring to autumn and winter) changes

in the Kilic19 and INESOSIM-PF snow depth estimates are consistent, and the monthly average INESOSIM-PF-estimated snow depth is close to the NSIDC snow depth.

## Data availability

ERA5 data is provided by the ECMWF and can be downloaded from the website (https://cds.climate.copernicus.eu). Brightness temperatures are available from the website (https://nsidc.org/data/AU_SI25/versions/1) and sea ice

concentrations are available from the website (https://nsidc.org/data/g02202/versions/3, https://nsidc.org/data/G10016/versions/1). Sea ice type and sea ice drift data are provided by the OSI SAF and can be downloaded from http://www.osi-saf.org/?q=content/global-sea-ice-type-c and http://www.osi-saf.org/?q=content/global-low-resolution-sea-ice-drift-c, respectively. Snow depth product and OIB data can be download from https://nsidc.org/data/RDEFT4/versions/1 and https://nsidc.org/data/NSIDC-0708/versions/1, respectively. IMB



measurements are provided by the CRREL (http://imb-crrel-dartmouth.org) and MOSAiC snow buoys data are provided by the AWI (https://data.meereisportal.de).

**Author contributions**

Haili Li developed the related snow depth retrieval method and undertook data processing. Qinghui Zhu compared different snow depth products. Chang-Qing Ke and Xiaoyi Shen offered suggestions to revised the manuscript.

**Competing interests**

The authors declare that they have no conflict of interest

**Acknowledgments**

This work was supported by the National Key Research and Development Program of China (grant no. 2018YFC1407200, no. 2018YFC1407203) and also supported by the National Natural Science Foundation of China (grant no. 41976212).

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
