# Peer review of "Estimating snow depth on Arctic sea ice based on reanalysis reconstruction and particle filter assimilation"

_The Cryosphere, 2021_

## Referee Comment (RC2)

Review of '**Estimating snow depth on Arctic sea ice based on reanalysis reconstruction and particle filter assimilation**' by Li et al.,

Review by Alek Petty

**Summary**

This study presents estimates of snow depth on Arctic sea ice from an updated version of the NASA Eulerian Snow on Sea Ice Model (NESOSIM) and a particle filter data assimilation scheme to combine the model estimates with satellite-derived snow depth data (RA-5VLSTM). The results were compared primarily with snow depths collected by NASA's Operation IceBridge and also some more limited Ice Mass Balance, and a MOSAiC snow depth buoys. The results were also compared with a Kilic et al., (2019) snow depth dataset produced from regression of IMBs to passive microwave data, and also the modified version of the Warren climatology.

**General comments**

In general, I think the approach of this study was good – use a new data assimilation approach to constrain NESOSIM output and potentially improve its ability to simulate snow depth on Arctic sea ice. However, I have a number of concerns about this study which I detail here:

1. NESOSIM is an open-source model (https://github.com/akpetty/NESOSIM) so community development is actively encouraged - e.g. adding new parameterizations, data assimilation modules etc., into the official code base. The framing of an Improved NESOSIM was thus slightly odd, although obviously this could also be a language/communication issue. The 'Improved' nature of this model framework was also somewhat underwhelming. The atmosphere loss term included as 1 of only 2 'improvements' in this version of NESOSIM has already been integrated into NESOSIM (v.1.1, https://github.com/akpetty/NESOSIM/releases/tag/v1.1). The authors made a note of this term being introduced already but made no link to the official code repo and still included it in your own 'improved' version. This means the only new parameterization introduced here (to generate the Improved NESOSIM framework) was the simple degree day temperature/melt parameterization. I think this parameter inclusion makes broad sense (we've considered something along these lines ourselves) but i) it was not actually clear that this specific parameterization helped improve the simulation of snow depth as most of the validation occurred in winter/spring and ii) this could have been communicated as a simple added parameter to NESOSIM. I think the atmosphere loss term was much more significant and we've found this to be a useful additional tuning factor, although one not well constrained by observations. Indeed most of what this study is doing is bias correcting towards the OIB quick-look data. On that note, I didn't see any information about making the code available (e.g. the degree day melt model or the particle assimilation approach) which was surprising considering the authors utilized extensively an open-source model for much of this work.

2. A big issue is that quick-look OIB snow depths are used as truth, with bias corrections/model calibration carried out to improve the fit to this dataset, essentially. However, deriving snow depths from Snow Radar data collected by OIB is challenging (Kwok et al., 2017,) and wide differences exist across the different products. We make a big point about this in the original NESOSIM paper (Petty et al., 2018, P2018). More recent research has shown that OIB QL is ~5 cm thinner than the consensus from the three 'final' products analyzed in P2018 (Petty et al., in prep), see preliminary figure below. These are (since 2013) quick-look data, supposed to provide a basic overview of sea ice conditions, not really a reliable dataset for validating models/retrievals.

[Figure]

*Figure 1: Comparison of the median snow depth from the three different OIB snow depth products used in Petty et al., (2018) and the quick-look (QL) OIB snow depth data. Data are gridded to a 100 km polar stereographic domain before the comparison.*

There were also plenty of other parts of the study where data uncertainties are vaguely described and, in some cases, described with worrying levels of certainty ('The satellite-derived snow depth contains an uncertainty of 1 cm,').

3. I was hoping this paper would provide a much deeper explanation and insight into particle filter data assimilation, but the paper provided only really a minimal description of this. In no way is the approach reproducible. It also left me feeling unsure how much the authors understood about the approach and how best to implement this. The particle number sensitivity test did not feel satisfactory.

4. The RA-5VLSTM dataset was used as the only input to the data assimilation system but the citation linked to is just a data portal that I was unable to translate, so really there is no background to how this data was obtained and how well it agrees with other snow depth datasets that exist. My guess is that the INESOSIM-PF run tracks this observational dataset quite closely, but it's unclear if that's a good thing or not.

**Specific comments**

The statistics of RMSE and MAE include the bias – so really all the statistics presented are highly sensitive to the presence of a bias. Most of this study seemed to involve basic bias correction (which is somewhat understandable considering the large uncertainties in snow) but limits the impact of the results presented. Generally I think it is not a good idea to express RMSE/bias changes as percentages. Just stating the change in absolute terms is easier for the reader to assess.

L73-74: this particle filter methodology and motivation needs to be much better described.

'Section 3.2 Two snow depth retrieval methods' – why are these not in the data section? They are previous data not really created in this study - one 'retrieval' - the multi-linear regression to passive microwave data from Kilic et al., (2019) and then the Warren 1999 (W99) quadratic fit to in-situ snow depths.

It is also confusing that you use the W99 with snow depths halved over FYI as well, and refer to this as an NSIDC product (taken from the CryoSat-2 implementation of this). This was also used in P2018 and is typically referred to as the modified Warren climatology. This was referenced in P2018 (and I think was first introduced by Laxon et al., 2013). I don't think it should be referred to as an NSIDC product particularly.

L201: 'However, the updated algorithm has not been debugged' is a bit of a strange way of framing this. The code is on GitHub (version 1.1) so you should ideally cite that more clearly, as it is exactly the same as the 'improved' atmosphere snow loss term used in this study.

L240 – 'Therefore, the model was insensitive to β.' – really, it's just insensitive in the regions where we have observations (e.g. in the central Arctic). P2018 showed how in more marginal seas it has a bigger impact.

L243 – now a bit confused regarding the parameter you're looking at here. I think it's beta but why choose that if NESOSIM is *less* sensitive to this parameter?

Section 4.1.1. – the problem here is that you're fitting to quick-look OIB now depths that are likely biased. To accommodate the product uncertainty in P2018 we looked at the different algorithms and noted the wide-spread made it hard to calibrate.

Figure 3 – this is not really a great way of showing differences/biases between runs as the lines all look basically the same.

Figure 4 - Bimodal NESOSIM output is interesting, what's going on there? I think P2018 showed weak evidence of bimodality.

L280-287 – but you don't seem to use the F labelling in the figures/tables?

Section 4.2 – these descriptions were generally quite unclear

'superiority of the assimilation results '

L353 – 355: 'We obtain the error in the Kilic19 snow depth based on the OIB-measured snow depth from 2018 to 2019.' This is a very bad idea!

Section 4.3 seemed superfluous. You compared against one snow depth dataset and the modified Warren climatology but without much context to guide this.

Figure 8a - I think something odd is happening in Figure 8a for that big start of October jump in snow depth. This needs to be looked into.

L444-445: 'The satellite-derived snow depth contains an uncertainty of 1 cm, and the NESOSIM snow depth uncertainty 445 reaches 5 cm (Petty et al., 2020).' Not sure where this is from. An uncertainty of 1 cm on what I assume are your snow depth measurements can't be right.

**References**

Kwok, R., Kurtz, N. T., Brucker, L., Ivanoff, A., Newman, T., Farrell, S. L., King, J., Howell, S., Webster, M. A., Paden, J., Leuschen, C., MacGregor, J. A., Richter-Menge, J., Harbeck, J., and Tschudi, M.: Intercomparison of snow depth retrievals over Arctic sea ice from radar data acquired by Operation IceBridge, The Cryosphere, 11, 2571–2593, https://doi.org/10.5194/tc-11-2571-2017, 2017.

Laxon, S. W., Giles, K. A., Ridout, A. L., Wingham, D. J., Willatt, R., Cullen, R., Kwok, R., Schweiger, A., Zhang, J., Haas, C., Hendricks, S., Krishfield, R., Kurtz, N., Far- rell, S., and Davidson, M.: CryoSat-2 estimates of Arctic sea ice thickness and volume, Geophys. Res. Lett., 40, 732–737, https://doi.org/10.1002/grl.50193, 2013.

Petty, A. A., M. Webster, L. N. Boisvert, T. Markus (2018), The NASA Eulerian Snow on Sea Ice Model (NESOSIM) v1.0: Initial model development and analysis, Geosci. Model Dev., doi: 10.5194/gmd-11-4577-2018.

---

## Author Comment (AC1)

**Response to Reviewers' Comments**

Dear reviewer,

We are grateful to receive your valuable and constructive comments in helping us improve this manuscript. According to your comments and suggestions, we have revised the manuscript seriously, including data, algorithm, discussion and conclusion. Please find the point-to-point responses as follows (Reviewer's comments in black and responses in blue). Thank you very much!

**Reviewer: 1**

Review on "Estimating snow depth on Arctic sea ice based on reanalysis reconstruction and particle filter assimilation" by Li et al.

The author provides a new method for estimating snow depth, and gives a detailed evaluation of data accuracy. Snow thickness is an important parameter in the cryosphere, which is of great significance to the mass balance of Arctic sea ice, the radiation balance of ocean and the retrieval of sea ice thickness using satellite altimeter data. Therefore, it is a paper suitable for publication in TC. However, at present, there are still many issues that need to be improved or corrected in the method and expression of this paper. Therefore, I recommend that the publication of the paper be considered after major revision.

General comments

The data used for validation is the focus of evaluating the product quality of the estimated snow thickness data. Therefore, it is necessary to evaluate the quality of the validation data itself and the characteristics of the data source in detail. For example, the data of buoys get the snow thickness on flat ice, which is generally low. A negative value does not indicate error, but indicates that the sea ice surface has melted, etc.

**Response:** Thank you for this thought-provoking suggestion. In the revised manuscript, we have added detailed information about the quality of the validation data and the characteristics of the data source.

(1) For the IMB data, the added contents are as follows:

The quality control of snow depth is applied and snow depth within 0–2 m is retained (Perovich et al., 2021). The IMB, equipped with an acoustic sounder, can measure the positions of the snow surface. The errors of IMB snow depth are within ±1 cm (Blanchard-wrigglesworth et al., 2018).

References:

Perovich, D., Richter-Menge, J., and Polashenski, C.: Observing and understanding climate change: Monitoring the mass balance motion, and thickness of Arctic sea ice, CRREL-Dartmouth Mass Balance Buoy Program, http://imb-crrel-dartmouth.org/, last access: 20 November 2021.

Blanchard-Wrigglesworth, E., Webster, M.A., Farrell, S.L., and Bitz, C.M.: Reconstruction of Snow on Arctic Sea Ice. J. Geophys. Res. Oceans, 123(5), 3588-3602, https://doi.org/10.1002/2017JC013364, 2018.

(2) For the OIB data, we have added two additional OIB products and the added contents are as follows:

The Operation IceBridge (OIB) mission is proposed for filling the data gap between ICESat and ICESat-2, providing snow depth on sea ice, sea ice thickness, and sea ice type information in the Arctic. These data are widely applied to evaluate satellite-derived or simulated snow depth values. In this study, three OIB products are used which are available to the public: (i) the IceBridge Sea Ice Freeboard, Snow Depth, and Thickness Quick Look, Version 1 (hereafter referred to as $OIB_{QL}$), covering the period 2012-2019; (ii) the IceBridge L4 Sea Ice Freeboard, Snow Depth, and Thickness, Version 1 (IDCSI4, hereafter referred to as $OIB_{IDCSI4}$), covering the period 2009-2013; and (iii) the Snow Depth on Arctic Sea Ice Data Set (Newman et al., 2014) which is provided by the NOAA (hereafter referred to as $OIB_{NOAA}$), covering the periods 2009-2012 and 2014-2015. $OIB_{QL}$ has a mean bias of about -5 cm, underestimating snow depth (Kwok et al., 2017). $OIB_{IDCSI4}$ product tends to underestimate snow depth (mean bias is about -1 cm) and $OIB_{NOAA}$ tends to overestimate snow depth (mean bias is about 2 cm).

References:

Newman, T., Farrell, S. L., Richter-Menge, J., Elder, B., Connor, L., Kurtz, N., and McAdoo, D.: Assessment of radar-derived snow depth measurements over Arctic sea ice. J. Geophys. Res.: Oceans, 119, 8578-8602, https://doi.org/10.1002/2014JC010284, 2014.

Kwok, R., Kurtz, N. T., Brucker, L., Ivanoff, A., Newman, T., Farrell, S. L., King, J., Howell, S., Webster, M. A., Paden, J., Leuschen, C., MacGregor, J. A., Richter-Menge, J., Harbeck, J., and Tschudi, M.: Intercomparison of snow depth retrievals over Arctic sea ice from radar data acquired by Operation IceBridge. The Cryosphere, 11(6), 2571-2593. https://doi.org/10.5194/tc-11-2571-2017, 2017.

(3) For the MOSAiC data, the added contents are as follows:

Snow buoys include four independent sonar measurements. A negative value does not indicate the error, but indicates that the sea ice surface has melted (Nicolaus et al., 2021).

Reference:

Nicolaus, M., Hoppmann, M., Lei, R., Belter, H. J., Fang, Ying-Chih., Rohde, J.: Snow height on sea ice, meteorological conditions and drift of sea ice from autonomous Snow Buoys

during               MOSAiC               2019/20.               PANGAEA, https://doi.pangaea.de/10.1594/PANGAEA.933742 (dataset in review), 2021.

This is a paper that introduces new methods and new data. The access path of new data should be given in the data availability section.

**Response:** According to the suggestion, we have uploaded the data to the National Tibetan Plateau Data Center. The data is now available at http://data.tpdc.ac.cn/en/disallow/5f33769e-8cd9-400e-ab2b-1b75657bec9f/.

Specific comments

Line 30: "limits solar radiation absorption" changes to "limits solar radiation absorption by the ocean".

**Response:** According to the suggestion, we have changed "limits solar radiation absorption" to "limits solar radiation absorption by the ocean".

Line 33: "Meltwater originating from thin snow" not just from thin snow, so, changes to "snow and ice surface"

**Response:** According to the suggestion, we have changed "Meltwater originating from thin snow" to "Meltwater originating from thin snow and ice surface".

2 Data: Instead of just listing data, we should give application purposes of different data at the beginning, which will make readers more understand the research ideas.

**Response:** Thank you for this thought-provoking suggestion. In the revised manuscript, we have given application purposes of different data at the beginning of each data introduction.

3 Line 100 "Data pertaining to the ten subregions covering the period from 2012–2020 are selected" --This sentence has been repeated several times.

**Response:** Thank you for this thought-provoking suggestion. In the revised manuscript, we have deleted the repeated sentences in sections 2.2 and 2.3.

4 Ice mass balance buoy (IMB) data are retrieved from the Cold Regions Research and Engineering Laboratory (CRREL)-Mass Balance Buoy Program-- This data base is initiated by the CRREL, but is jointly maintenanced by the CRREL and University of Dartmouth.

**Response:** Thank you for this thought-provoking suggestion. In the revised manuscript, we have changed "Ice mass balance buoy (IMB) data are retrieved from the Cold Regions Research and Engineering Laboratory (CRREL)-Mass Balance Buoy Program" to "IMB data base is

initiated by the Cold Regions Research and Engineering Laboratory (CRREL), but is jointly maintenanced by the CRREL and University of Dartmouth".

5 Line 125 "This dataset is developed to monitor the sea ice volume": The IMB cannot monitor the ice volume because it is not the point measurement.

**Response:** We extremely agree with this suggestion. In the revised manuscript, we have changed "This dataset is developed to monitor the sea ice volume and mass balance to better understand climate change" to "This dataset is developed to monitor the mass balance of the sea ice cover to better understand climate change".

6 Line 170: Blowing snow lost to leads: wind forcing causes any snow lost from the new snow layer to lead/open water: When the sea ice is relatively compact, the destination of wind blown snow may not be in the waterway or open water, but also in the downwind direction of the ice ridge. Therefore, the snow depth of level ice is generally smaller than that over the ridge.

Refer to:

Wagner, DN, Shupe, MD, Persson, OG, Uttal, T, Frey, MM, Kirchgaessner, A, Schneebeli, M, Jaggi, M, Macfarlane, AR, Itkin, P, Arndt, S, Hendricks, S, Krampe, D, Ricker, R, Regnery, J, Kolabutin, N, Shimanshuck, E, Oggier, M, Raphael, I, Lehning, M. 2021. Snowfall and snow accumulation processes during the MOSAiC winter and spring season. The Cryosphere Discussions: 1-48. DOI: https://doi.org/10.5194/tc-2021-126.

Lei R*, Tian-Kunze X, Li B, Heil P, Wang J, Zeng J, Tian Z. 2017. Characterization of summer Arctic sea ice morphology in the 135°-175°W sector using multi-scale methods, Cold Regions Science and Technology, 133, 108–120.

**Response:** Thank you for this thought-provoking suggestion. We agree that the destination of wind-blown snow may not only be in the lead/open water, but also in the downwind direction of the ice ridge. The roughness of the ice ridge is large, and the wind-blown snow is hindered by the ice ridge and distributed around the ice ridge. Therefore, the snow depth of level ice is generally smaller than that over the ridge. Ice ridges are mainly distributed in multi-year ice areas, and they have an impact on the distribution of snow depth in local areas. Parametric processes considered in this paper will affect the snow depth in the whole Arctic, so the influence is greater than that of the ice ridge. Although the blowing snow lost to ice ridge is not considered in the parameterization process of the model at present, the final snow depth estimates assimilate the satellite-derived snow depth and can capture the high snow depths over the ice ridge, which weakens the impact caused by the lack of consideration of ice ridge in NESOSIM_M model. In the future, we can deeply study the influence of ice ridge over snow depth and try to parameterize the process of blowing snow lost to ice ridge and introduce it into the numerical model, so as to improve the model. In the revised manuscript, we have added the

limitations of the parameterization process in section 5 and proved that the assimilated snow depth obtained in this paper had captured the high snow depth over the ice ridge, indicating the advantages of the model after the assimilation of satellite-derived snow depth.

(1) In section 2.5, we added the introduction of Landsat 8:

Landsat 8 was launched on February 11, 2013, and was equipped with the Operational Land Imager (OLI) and Thermal Infrared Sensor (TIRS). OLI includes nine bands. Except for the panchromatic band with a spatial resolution of 15 m, the remaining eight bands have a spatial resolution of 30 m. Due to its high resolution, it can find finer details on sea ice. Therefore, Landsat 8 Level 1 Terrain Corrected (L1T) product is downloaded to distinguish the ice ridges.

(2) In section 5.2, we added the related content of ice ridges according to the Landsat 8:

Based on Landsat 8 images, it was found that ice ridges were distributed in the Beaufort Sea on March 22, 2020, while there were no ridges in the same area on March 6 (Fig. 11(a) and (b)), indicating that ice ridges were generated from March 6 to March 22. NESOSIM_M snow depth on March 22 was greater than on March 6, but NESOSIM_M snow depth decreased from March 15 to March 22 (Fig. 11(c)). The existence of an ice ridge will promote snow accumulation, and the snow depth over the ice ridge is higher than that of level ice (Lei et al., 2017; Wagner et al., 2021). The NESOSIM_M ignored the influence of the existence of ice ridge on snow depth. Meanwhile, the change of NESOSIM_M-PF snow depth well reflected the increase of snow depth over the ice ridge (Fig. 11(c)). Point A represented the NESOSIM_M-PF snow depth on March 6 (i.e., 11.3 cm), and point B represented the NESOSIM_M-PF snow depth on March 22 (i.e., 16.0 cm), with an increase of 4.7 cm. From March 6 to March 22, NESOSIM_M-PF snow depth increased. The snow depth increased rapidly from March 17 and reached the local maximum on March 22. It proves that the NESOSIM_M-PF can capture the high snow depths over the ice ridge and weaken the impact caused by the lack of consideration of ice ridge in the NESOSIM_M. In the future, we will try to parameterize the process of blowing snow lost to ice ridge and introduce it into the numerical model to improve the NESOSIM_M.

[Figure]

**Fig 11.** Distribution of the ice ridge in the Beaufort Sea on (a) March 6, 2020, and (b) March 22, 2020. (c) Variations in NESOSIM_M-PF and NESOSIM_M snow depth in the red box (the box is shown in (a) and (b)) in March 2020. Note that the blue arrow of (b) shows the location of ice ridges, point A and B in (c) represent the NESOSIM_M-PF snow depth on March 6 and March 22, respectively.

References:

Lei, R., Tian-Kunze, X., Li, B., Heil, P., Wang, J., Zeng, J., Tian, Z.: Characterization of summer Arctic sea ice morphology in the 135°-175°W sector using multi-scale methods, Cold Reg. SCI. Technol., 133, 108-120, https://doi.org/10.1016/j.coldregions.2016.10.009, 2017.

Wagner, D. N., Shupe, M. D., Persson, O. G., Uttal, T., Frey, M.M., Kirchgaessner, A., Schneebeli, M., Jaggi, M., Macfarlane, A. R., Itkin, P., Arndt, S., Hendricks, S., Krampe, D., Ricker, R., Regnery, J., Kolabutin, N., Shimanshuck, E., Oggier, M., Raphael, I., Lehning, M.: Snowfall and snow accumulation processes during the MOSAiC winter and spring season. The Cryosphere Discussions: 1-48, DOI: https://doi.org/10.5194/tc-2021-126, 2021.

7 Line 195 "the 2-m temperature (Tair) is higher than 0 ℃": Snow may also melt below 0 degrees Celsius, mainly due to solar radiation.

Refer to:

Bliss, A. C., & Anderson, M. R. (2018). Arctic Sea Ice Melt Onset Timing From Passive Microwave-Based and Surface Air Temperature-Based Methods. Journal of Geophysical Research: Atmospheres, 123(17), 9063-9080.

**Response:** Thank you for this thought-provoking suggestion. We agree that snow may also melt below 0 ℃, mainly due to solar radiation. Different studies have adopted different thresholds to determine the melt onset (MO). Bliss and Anderson (2018) compared the MO obtained by three different thresholds. They found that MO using 0 ℃ or thresholds from Rigor et al. (2000) was later than that using the threshold of -1 ℃. There are also studies using -0.5 ℃ (Lindsay, 1998) and -1.9 ℃ (Andreas and Ackley, 1982) as thresholds. Under different circumstances, such as changes in salinity will change the melting point of snow. We use 0 ℃ as the threshold to judge the existence of the melting term, which will inevitably underestimate the snow loss caused by snow melting. The model starts running in mid-August because there is heavy snowfall in the central Arctic in August, and great snow melting events in June and July have been avoided (Petty et al., 2018). In mid-August, sea ice is mainly distributed in the central Arctic, and snow melting events also mainly occur in the central Arctic. Stroeve et al. (2006) revealed that the MO determined by the threshold closer to 0 ℃ would agree more closely with passive microwave (PMW)-based MO dates for the sea ice within the central Arctic. Therefore, we choose 0 ℃ as the threshold, assuming that when the air temperature is less than 0 ℃, the snow melting term is negligible. According to the suggestion, we have added references to explain why we chose 0 ℃ as the threshold and discussed limitations for the current snow melting process.

(1) In section 3.2.2, we added reasons for selecting the threshold of 0 ℃ as follows:

The mid-August is selected because there is heavy snowfall in the central Arctic, and great snow melting events in June and July have been avoided (Petty et al., 2018). In mid-August, sea ice is mainly distributed in the central Arctic, and snow melting events also mainly occur in the central Arctic. For the sea ice within the central Arctic, Stroeve et al. (2006) revealed that the melt onset (MO) determined by the threshold closer to 0 ℃ would agree more closely with passive microwave (PMW)-based MO dates. Therefore, we choose 0 ℃ as the threshold. When the 2-m temperature ($T_{air}$) is higher than 0 ℃, we consider that there occurs a snow melting process on sea ice.

(2) In section 5.2, we added limitations for selecting the threshold of 0 ℃ as follows:

Moreover, snow may also melt below 0 ℃, mainly due to solar radiation. Different studies have adopted different thresholds to determine the MO. Bliss and Anderson (2018) compared the MO obtained by three different thresholds, and found that MO using 0 ℃ or thresholds from Rigor et al. (2000) was later than that using the threshold of -1 ℃. There are also studies using -0.5 ℃ (Lindsay, 1998) and -1.9 ℃ (Andreas and Ackley, 1982) as thresholds. Under different circumstances, the melting point of snow is not fixed. We use 0 ℃ as the threshold to

judge the existence of the melting term, which will inevitably underestimate the snow loss caused by snow melting.

References:

Andreas, E. L., and Ackley, S. F.: On the differences in ablation seasons of Arctic and Antarctic Sea ice. J. Atmos. Sci., 39(2), 440-447, https://doi.org/10.1175/1520-0469(1982)039<0440:OTDIAS>2.0.CO;2, 1982.

Bliss, A. C., and Anderson, M. R.: Arctic Sea Ice Melt Onset Timing From Passive Microwave-Based and Surface Air Temperature-Based Methods. J. Geophys. Res.: Atmospheres, 123(17), 9063-9080, https://doi.org/10.1029/2018JD028676, 2018.

Lindsay, R. W.: Temporal variability of the energy balance of thick arctic pack ice. J. Climate, 11(3), 313-333, https://doi.org/10.1175/1520-0442(1998)011<0313:TVOTEB>2.0.CO;2, 1998.

Rigor, I. G., Colony, R. L., and Martin, S.: Variations in surface air temperature observations in the Arctic. J. Climate, 13(5), 896-914, https://doi.org/10.1175/1520-0442(2000)013<0896:VISATO>2.0.CO;2, 2000.

Stroeve, J., Markus, T., Meier, W. N., and Miller, J.: Recent changes in the Arctic melt season. Ann. Glaciol., 44, 367-374, https://doi.org/10.3189/172756406781811583, 2006.

8 Line 200 "wind transports snow into the atmosphere" Most of the snow due to the Blowing snow will fall back to the ice, but there is a spatial redistribution. Main mechanism to transport snow into the atmosphere is evaporation.

**Response:** Thank you for this thought-provoking suggestion. Indeed, most of the snow due to the Blowing snow will fall back to the ice. The main mechanism to transport snow into the atmosphere is evaporation. However, for a grid cell, the wind will bring snow to the atmosphere, causing the redistribution of snow and changing the snow depth of the grid cell. In 2020, Petty (2020) proposed that the snow lost to the atmosphere process when wind speed exceeds the threshold should be considered in NESOSIM. In this study, we do not consider the snow loss to the atmosphere caused by evaporation, but only the blowing snow loss to the atmosphere by the wind. In the revised manuscript, we have added this limitation of the proposed method in the discussion.

The added contents are as follows:

Furthermore, the NESOSIM_M-PF only considers the simple melting process and does not involve the snow loss caused by evaporation. More complex thermodynamics processes need to be further considered in the future.

9 Line 243 "OIB-measured snow depth is 10.79 cm": Whether the two digits after the decimal point are meaningful? also in other similar places. According to my understanding, the observation accuracy of snow depth can hardly be better than 1cm.

**Response:** Thank you for this thought-provoking suggestion. Some studies use values that

include two digits after the decimal point (e.g., Rostosky et al., 2018; Kwok et al., 2017) to describe snow depth. Some studies use one digit after the decimal point (e.g., Kilic et al., 2019; Zhou et al., 2021; Stroeve et al., 2020), and some studies do not retain the number after the decimal point (Petty et al., 2018). In this paper, NESOSIM_M-PF snow depth was obtained, and the errors of different snow depths were compared. When the improvement of accuracy is small, the improvement of accuracy will be ignored if we use no digits after the decimal. For example, if we use no digits after the decimal, the RMSE of the two snow depth estimates (NESOSIM v1.0 and NESOSIM_M snow depths) is 7 cm (Based on the $OIB_{QL}$ data). However, the RMSE of NESOSIM_M is less than 7 cm, and the RMSE of NESOSIM v1.0 is greater than 7 cm. Keeping one and two decimal places will have less impact on the results of this study. According to the suggestion and references, we chose the commonly used strategies to describe snow depth (i.e., one digit after the decimal point). Referring to Kwok et al. (2017) and Stroeve et al. (2020), we changed the description in the manuscript to use one digit after the decimal point and the values in the table and figures still retained two decimal places.

10 Line 283 "the IMB-measured snow depth are much smaller than 0, indicating great snow depth underestimation": it is not underestimation, but means the melt of ice surface.

**Response:** We are sorry that this sentence caused a misunderstanding. This sentence means the biases between three estimated snow depths (F1, F2 and F3 snow depth) and the IMB-measured snow depths are less than 0, rather than the IMB-measured snow depth are smaller than 0. In the revised manuscript, we have changed "The biases between the F1, F2 and F3 snow depths and the IMB-measured snow depth are much smaller than 0, indicating great snow depth underestimation" to "The negative biases indicate these three schemes (i.e., FI, F2 and F3) underestimate the snow depth (Table 1)".

**Table 1.** Accuracy of NESOSIM_M with different atmospheric loss coefficient values ($\gamma$) based on the IMB-measured snow depth (number of same matching points (Ns), RMSE (cm), bias (cm), MAE (cm) and r).

| $\gamma$ | 0.015 (F1) | 0.020 (F2) | 0.025 (F3) |
|---|---|---|---|
| Ns | 443 | 443 | 443 |
| RMSE (cm) | 16.58 | 16.85 | 17.11 |
| Bias (cm) | -6.67 | -7.36 | -7.97 |
| MAE (cm) | 11.06 | 11.28 | 11.51 |
| r | 0.12 | 0.12 | 0.11 |

11 Figure 8: what the meaning fro the increased jump at the end of September?

**Response:** Thank you for this thought-provoking suggestion. Except for August and September, the satellite-derived snow depth has been used for assimilation. There is no satellite-derived snow depth in August and September. Therefore, the estimated snow depth in August and September is the NESOSIM_M snow depth, resulting in the increased jump at the end of

September.

We are sorry we ignored this increased jump earlier. To solve the increased jump at the end of September, we use the NESOSIM_M-PF snow depth and NESOSIM_M snow depth at the same time and location in October to establish the linear regression equation as follows:

$$h_{NESOSIM\_M\text{-}PF}=1.2138\times h_{NESOSIM\_M}+0.9214 \qquad (21)$$

We use Eq. (21) to obtain NESOSIM_M-PF snow depths in August and September. The results show that the increased jump at the end of September disappears and variation in snow depth from September to May is more reasonable (Fig. AA).

[Figure]

Figure AA. (a) Variations in the daily average Arctic default snow depth (no post-processing). (b) Variations in the daily average Arctic default snow depth (with post-processing).

In the revised manuscript, we have added the additional processing for eliminating the increased jump at the end of September. The revisions are as follows:

Except for August and September, the satellite-derived snow depth has been used for assimilation. There is no satellite-derived snow depth in August and September. Therefore, the estimated snow depth in August and September is the NESOSIM_M snow depth, resulting in the increased jump at the end of September. To solve this problem, we use the NESOSIM_M-PF snow depth and NESOSIM_M snow depth at the same time and location in October to

establish the linear regression equation as follows:

$$h_{NESOSIM\_M\text{-}PF}=1.2138 \times h_{NESOSIM\_M}+0.9214 \tag{21}$$

We use Eq. (21) to obtain NESOSIM_M-PF snow depths in August and September.

---

## Author Comment (AC2)

**Response to Reviewers' Comments**

**Dear Petty,**

We are grateful to receive your valuable and constructive comments in helping us improve this manuscript. According to your comments and suggestions, we have revised the manuscript seriously, including data, algorithm, discussion and conclusion. Please find the point-to-point responses as follows (Reviewer's comments in black and responses in blue). Thank you very much!

**Reviewer: 2**

Review of 'Estimating snow depth on Arctic sea ice based on reanalysis reconstruction and particle filter assimilation' by Li et al.,

Review by Alek Petty

**Summary**

This study presents estimates of snow depth on Arctic sea ice from an updated version of the NASA Eulerian Snow on Sea Ice Model (NESOSIM) and a particle filter data assimilation scheme to combine the model estimates with satellite-derived snow depth data (RA-5VLSTM). The results were compared primarily with snow depths collected by NASA's Operation IceBridge and also some more limited Ice Mass Balance, and a MOSAiC snow depth buoys. The results were also compared with a Kilic et al., (2019) snow depth dataset produced from regression of IMBs to passive microwave data, and also the modified version of the Warren climatology.

General comments

In general, I think the approach of this study was good–use a new data assimilation approach to constrain NESOSIM output and potentially improve its ability to simulate snow depth on Arctic sea ice. However, I have a number of concerns about this study which I detail here:

1. NESOSIM is an open-source model (https://github.com/akpetty/NESOSIM) so community development is actively encouraged - e.g., adding new parameterizations, data assimilation modules etc., into the official code base. The framing of an Improved NESOSIM was thus slightly odd, although obviously this could also be a language/communication issue. The 'Improved' nature of this model framework was also somewhat underwhelming. The atmosphere loss term included as 1 of only 2 'improvements' in this version of NESOSIM has already been integrated into NESOSIM (v.1.1, https://github.com/akpetty/NESOSIM/releases/tag/v1.1). The

authors made a note of this term being introduced already but made no link to the official code repo and still included it in your own 'improved' version. This means the only new parameterization introduced here (to generate the Improved NESOSIM framework) was the simple degree day temperature/melt parameterization. I think this parameter inclusion makes broad sense (we've considered something along these lines ourselves) but i) it was not actually clear that this specific parameterization helped improve the simulation of snow depth as most of the validation occurred in winter/spring and ii) this could have been communicated as a simple added parameter to NESOSIM. I think the atmosphere loss term was much more significant and we've found this to be a useful additional tuning factor, although one not well constrained by observations. Indeed most of what this study is doing is bias correcting towards the OIB quicklook data. On that note, I didn't see any information about making the code available (e.g., the degree day melt model or the particle assimilation approach) which was surprising considering the authors utilized extensively an open-source model for much of this work.

**Response:** Thank you for this thought-provoking suggestion. We are sorry for the misunderstanding caused by the description in the initial manuscript. We redescribe the method in this paper. The atmospheric loss term, proposed by Petty (2020), makes broad sense. However, the melting term is also an essential process. With global warming, the melting process will be more intense and its contribution to the change of snow depth will increase. The melting process we currently consider is relatively simple. In the future, we will consider more complex melting processes to continuously develop the model.

(1) According to the suggestion, we rename the methods in this paper as NESOSIM v1.0 (including four parameterization processes), NESOSIM\_M (adding additional atmospheric loss term and melting term, M refers to melting process) and NESOSIM\_M-PF (Data assimilation is included).

(2) Then, we give the website of official code, https://github.com/akpetty/NESOSIM, indicating that the NESOSIM is an open-source model. The detailed revisions are as follows:

Petty et al. (2018) developed a two-layer snow depth model (i.e., NESOSIM v1.0), and snow was divided into a new snow layer and an old layer. NESOSIM is an open-source model (https://github.com/akpetty/NESOSIM) that offers public for contributing their efforts to further develop this model.

(3) Finally, when describing the NESOSIM\_M model, we point out that the atmospheric loss term was proposed by Petty (2020). On this basis, we propose that the melting term needs to be considered in this model. The detailed revisions are as follows:

**3.2.2 NESOSIM\_M**

In addition to wind packing and blowing snow loss to leads, wind cause snow loss to the atmosphere, resulting in the redistribution in snow depth. In 2020, Petty (2020) updated the

NESOSIM v1.0 and proposed that the snow lost to the atmosphere process should be considered. Similar to the blowing snow lost to leads and wind packing processes, snow is lost to the atmosphere when U exceeds 5 m s-1. The atmospheric loss term is determined by the blowing snow coefficient, atmospheric loss coefficient ( $\gamma$ ), wind speed and depth of the new snow layer. The equation is as follows:

$$\Delta h_s^{atm}(t) = -\beta \gamma UTh_s(t, 0)$$

(12)

Besides adding the snow lost to the atmosphere term proposed by Petty (2020), we introduce a simple melting term to develop the NESOSIM\_M.

With the continuous warming of the Arctic, snow melting becomes increasingly dramatic. In this study, NESOSIM starts to run in mid-August and continues to run until the following mid-May. The mid-August is selected because there is heavy snowfall in the central Arctic, and great snow melting events in June and July have been avoided (Petty et al., 2018). In mid-August, sea ice is mainly distributed in the central Arctic, and snow melting events also mainly occur in the central Arctic. For the sea ice within the central Arctic, Stroeve et al. (2006) revealed that a threshold closer to 0 °C would agree more closely with passive microwave (PMW)-based melt onset (MO) dates. Therefore, we choose 0 °C as the threshold. When the 2-m temperature ( $T_{air}$ ) is higher than 0 °C, we consider that there occurs a snow melting process on sea ice.

$$\Delta h_s^{melt}(t) = -T_{air}(t) T \tau \rho_w / \rho_s^n \tag{13}$$

where  $\tau$  is the degree-day factor and  $\rho_w$  is the water density. We set  $\tau$  to  $6.3 \times 10^{-8}$  m °C-1 s-1 (Kuchment and Gelfan, 1996), which is determined via the degree-day method.

2. A big issue is that quick-look OIB snow depths are used as truth, with bias corrections/model calibration carried out to improve the fit to this dataset, essentially. However, deriving snow depths from Snow Radar data collected by OIB is challenging (Kwok et al., 2017,) and wide differences exist across the different products. We make a big point about this in the original NESOSIM paper (Petty et al., 2018, P2018). More recent research has shown that OIB QL is ~5 cm thinner than the consensus from the three 'final' products analyzed in P2018 (Petty et al., in prep), see preliminary figure below. These are (since 2013) quick-look data, supposed to provide a basic overview of sea ice conditions, not really a reliable dataset for validating models/retrievals.

Figure 1: Comparison of the median snow depth from the three different OIB snow depth products used in Petty et al., (2018) and the quick-look (QL) OIB snow depth data. Data are gridded to a 100 km polar stereographic domain before the comparison.

**Response:** Thank you for this thought-provoking suggestion. We agree that OIB quick look (OIBOL) product has relatively large errors compared with other 'final' products. We are sorry that we do not describe the error of OIBQL clearly. At present, there are three OIB products available to the public: (i) the IceBridge Sea Ice Freeboard, Snow Depth, and Thickness Quick Look, Version 1 (hereafter referred to as  $OIB_{OL}$ ), covering the period 2012-2019; (ii) the IceBridge L4 Sea Ice Freeboard, Snow Depth, and Thickness, Version 1 (IDCSI4, hereafter referred to as OIBIDCSI4), covering the period 2009-2013; and (iii) the Snow Depth on Arctic Sea Ice Data Set (Newman et al., 2014) which is provided by the NOAA (hereafter referred to as OIBNOAA), covering the periods 2009-2012 and 2014-2015. In the current situation of scarce in-site data, the OIBOL product provides more data than the other two OIB products and can provide us with an intuitive comparison result of snow depth as well. However, before using this data, the error of the product should be clarified. The OIBOL product underestimates the snow depth, and the mean bias is about -5 cm. According to the suggestion, when determining the model, we not only use the OIBOL product to determine the model parameters, but also add the accuracy evaluations based on the OIBIDCSI4 product to further determine the model parameters.

In section 4.2, we added OIBNOAA and OIBIDCSI4 to evaluate snow depth estimates. The results show that after adding melting term and atmospheric loss term, the accuracy of NESOSIM\_M snow depth decreased, but the accuracy of the NESOSIM\_M-PF snow depth has been greatly improved compared to NESOSIM v1.0, NESOSIM\_M and RA-5VLSTM. When using the MOSAiC product to evaluate snow depth estimates, NESOSIM\_M snow depth accuracy is significantly better than NESOSIM v1.0 snow depth. The accuracy of NESOSIM\_M-PF snow depth is better than that of satellite-derived snow depth and NESOSIM\_M snow depth. In the future, more in-site data are needed to further evaluate the results, continuously optimize the model parameters and improve the snow depth estimates.

The detailed revisions are as follows:

(1) We have added two  $OIB_{IDCSI4}$  and  $OIB_{NOAA}$  products and more  $OIB_{QL}$  information in section 2.6 as follows:

The Operation IceBridge (OIB) mission is proposed for filling the data gap between ICESat and ICESat-2, providing snow depth on sea ice, sea ice thickness, and sea ice type information in the Arctic. These data are widely applied to evaluate satellite-derived or simulated snow depth values. In this study, three OIB products are used which are available to the public: (i) the IceBridge Sea Ice Freeboard, Snow Depth, and Thickness Quick Look, Version 1 (hereafter referred to as OIBQL), covering the period 2012-2019; (ii) the IceBridge L4 Sea Ice Freeboard, Snow Depth, and Thickness, Version 1 (IDCSI4, hereafter referred to as OIBIDCSI4), covering the period 2009-2013; and (iii) the Snow Depth on Arctic Sea Ice Data Set (Newman et al.,

2014) which is provided by the NOAA (hereafter referred to as  $OIB_{NOAA}$ ), covering the periods 2009-2012 and 2014-2015.  $OIB_{QL}$  has a mean bias of about -5 cm, underestimating snow depth (Kwok et al., 2017).  $OIB_{IDCSI4}$  product tends to underestimate snow depth (mean bias is about - 1 cm) and  $OIB_{NOAA}$  tends to overestimate snow depth (mean bias is about 2 cm).  $OIB_{QL}$  data from 2014 to 2017 are considered to develop the snow depth model (Fig. 2(a)).  $OIB_{QL}$  data from 2018 to 2019,  $OIB_{IDCSI4}$  in 2013 and  $OIB_{NOAA}$  from 2014 to 2015 are employed to evaluate the established snow depth model (Fig. 2(b), 2(c) and 2(d)).